# Embeddings as Probabilistic Equivalence in Logic Programs

**Jaron Maene**
KU Leuven
Leuven, Belgium
jaron.maene@kuleuven.be

**Efthymia Tsamoura**
Huawei Labs*
Cambridge, United Kingdom
efthymia.tsamoura@huawei.com

## Abstract

The integration of logic programs with embedding models resulted in a class of neurosymbolic frameworks that jointly learn symbolic rules and representations for the symbols in the logic (constant or predicate). The key idea that enabled this integration was the *differentiable relaxation of unification*, the algorithm for variable instantiation during inference in logic programs. Unlike unification, its relaxed counterpart exploits the similarity between symbols in the embedding space to decide when two symbols are semantically equivalent. We show that this similarity between symbols violates the transitive law of equivalence, leading to undesirable side effects in learning and inference. To alleviate those side effects, we are the first to revamp the well-known possible world semantics of probabilistic logic programs into new semantics called *equivalence semantics*. In our semantics, a probabilistic logic program induces a probability distribution over all possible equivalence relations between symbols, instead of a probability distribution over all possible subsets of probabilistic facts. We propose a factorization of the equivalence distribution using latent random variables and characterize its expressivity. Additionally, we propose both exact and approximate techniques for reasoning in our semantics. Experiments on well-known benchmarks show that the equivalence semantics leads to neurosymbolic models with up to 42% higher results than state-of-the-art baselines.

## 1 Introduction

*Probabilistic logic programs* offer a principled way to reason in the presence of uncertain knowledge [De Raedt and Kimmig, 2015]. In neurosymbolic AI, this uncertain knowledge is provided by neural models [Marra et al., 2024, Feldstein et al., 2024]. Probability can hence be the "glue" between symbolic background knowledge and neural networks, leading to end-to-end differentiable architectures [Rocktäschel and Riedel, 2017, Manhaeve et al., 2021].

The most straightforward way to reason symbolically over the outputs of neural networks is by treating them as probabilistic facts [Manhaeve et al., 2021]. A second alternative is to map the constants and predicates of the logic program into a representation space, achieving a tighter integration. The question hence arises, *how can we apply a symbolic rule over elements in an embedding space?* The answer is via adopting the simple, yet powerful notion of *soft unification* [Rocktäschel and Riedel, 2017], which relaxes the *unification* algorithm in logic programming [Sessa, 2002]. Instead of requiring two constants and predicates to be syntactically the same, they need to be "semantically equivalent" to some extent. We demonstrate the notion of unification and its soft counterpart below.

**Example 1.** *Consider the rule* $\mathsf{isIn}(X, Y) \leftarrow \mathsf{isIn}(X, Z) \wedge \mathsf{isIn}(Z, Y)$ *and the following three facts:* $\mathsf{locatedIn}(\mathsf{eiffel\_tower}, \mathsf{paris})$, $\mathsf{isIn}(\mathsf{paris}, \mathsf{france})$, $\mathsf{isIn}(\mathsf{france}, \mathsf{europe})$. *In the context of logic pro-*

---

*Work started before Efthymia Tsamoura joined Huawei Labs.

**Possible world semantics**     **Probabilistic equivalence semantics**

0.9 :: w.
0.1 :: b.
a ← b ∧ w.

w.
h.
a ← b ∧ w.

$p(\mathsf{w} \wedge \neg\mathsf{b}) = 0.81$     $p(\mathsf{w} \wedge \mathsf{b}) = 0.09$     $p\left(\begin{smallmatrix}\mathsf{w}\ \mathsf{h}\\\mathsf{a}\ \mathsf{b}\end{smallmatrix}\right) = 0.34$     $p\left(\begin{smallmatrix}\mathsf{w}\ \mathsf{h}\\\mathsf{a}\ \mathsf{b}\end{smallmatrix}\right) = 0.01$

w.

a ← b ∧ w.

w.
b.
a ← b ∧ w.

w.
h.
a ← b ∧ w.

a.
a.
a ← a ∧ a.

$p(\neg\mathsf{w} \wedge \neg\mathsf{b}) = 0.09$     $p(\neg\mathsf{w} \wedge \mathsf{b}) = 0.01$     $p\left(\begin{smallmatrix}\mathsf{w}\ \mathsf{h}\\\mathsf{a}\ \mathsf{b}\end{smallmatrix}\right) = 0.51$

a ← b ∧ w.

b.
a ← b ∧ w.

w.
b.
a ← b ∧ w.

$\cdots$

Figure 1: Example probabilistic logic programs, under the possible world (left) and the probabilistic equivalence semantics (right). The top row displays the probabilistic programs, while the rows below are the discrete programs in the corresponding program distributions. Above each different program, we write its probability according to the distribution.

*gramming, unification is the mechanism to derive new facts by establishing a mapping $\sigma$ from variables to constants. This mapping aims to instantiate the premise of a rule using the existing facts. In our example, applying the mapping $\sigma = \{\mathsf{X} \mapsto \mathsf{paris}, \mathsf{Y} \mapsto \mathsf{europe}, \mathsf{Z} \mapsto \mathsf{france}\}$ to our rule, we can derive the new fact $\mathsf{isIn}(\mathsf{paris}, \mathsf{europe})$ using the existing facts $\mathsf{isIn}(\mathsf{paris}, \mathsf{france})$ and $\mathsf{isIn}(\mathsf{france}, \mathsf{europe})$. Establishing $\sigma$ is only possible because $\mathsf{isIn}(\mathsf{paris}, \mathsf{france})$ shares the same predicate $\mathsf{isIn}$ with the atom $\mathsf{isIn}(X, Z)$ and the fact $\mathsf{isIn}(\mathsf{france}, \mathsf{europe})$ shares the same predicate with the atom $\mathsf{isIn}(Z, Y)$. The fact $\mathsf{isIn}(\mathsf{eiffel\_tower}, \mathsf{france})$ cannot be derived: unification fails as the predicate $\mathsf{locatedIn}$ is distinct from $\mathsf{isIn}$. To support such cases, soft unification extends unification by allowing us to consider as being the same symbols (constants or predicates) that are different from each other, yet semantically equivalent. In our example, soft unification does identify $\sigma$, subject to $\mathsf{isIn}$ being equivalent to $\mathsf{locatedIn}$. The probability that two symbols are equivalent is determined by their proximity in the underlying representation space.*

**Motivation.** Soft unification has been particularly appealing, as it enables simultaneously learning symbolic rules and symbol embeddings in an end-to-end differentiable fashion [Campero et al., 2018, Weber et al., 2019, Minervini et al., 2020a,b, Maene and De Raedt, 2023]. The common characteristic of all these techniques is that they all use a *similarity* between symbols, disregarding that equivalence is transitive. However, abusing the semantics of equivalence in probabilistic logic programming has problematic consequences in learning and inference. More precisely, we show that soft unification leaks probability mass and suffers from local optima.

**Contributions.** To overcome the above consequences, we revamp the semantics of probabilistic logic programs, so that they induce a distribution $P_{\mathcal{E}}$ over *all possible equivalence relations* rather than a distribution over *all possible worlds* (all possible subsets of facts in the program). Figure 1 visualizes how placing uncertainty on equivalence between constants compares to the usual possible world semantics. Our semantics, referred to as *probabilistic equivalence*, allows us to treat symbol embeddings as a factorization of $P_{\mathcal{E}}$ and to reason over them. Additionally, we re-purpose existing inference techniques developed for reasoning under equivalence for the probabilistic setting. We propose an exact inference pipeline based on singularization [Marnette, 2009] and the magic sets transformation [Beeri and Ramakrishnan, 1987], as well as a Monte-Carlo estimator for approximate inference. Our empirical analysis against state-of-the-art techniques that learn and reason over symbol embeddings relying on soft unification [Rocktäschel and Riedel, 2017, Minervini et al., 2020a, Maene and De Raedt, 2023] shows that our semantics leads to models with up to 42% higher accuracy over the state-of-the-art. In summary, we make the following contributions:

- In Section 3.1, we show that soft unification violates the semantics of equivalence in probabilistic logic programming and investigate its side-effects for learning and inference.
- We propose a sound distribution semantics over probabilistic logic programs under equivalence relations called equivalence semantics in Section 4. Our semantics overcomes the independence assumption in probabilistic logic programming [Van Krieken et al., 2024].
- In Section 4.1, we propose a factorization of the program distribution induced by our semantics using latent random variables and characterize its expressivity. In this factorization, we realize the latent random variables as symbol embeddings enabling reasoning over them.
- We introduce exact and approximate inference techniques for our semantics, in Section 5.
- We empirically assess our proposed technique in link prediction and structure learning and compare it against state-of-the-art neurosymbolic approaches that employ soft unification. Our results show that our equivalence semantics improves performance up to 42%.

Supplementary material, including all code and proofs, is available at `https://github.com/ML-KULeuven/equality_reasoning`.

## 2  Preliminaries

We first briefly introduce (probabilistic) logic programs and equivalence relations. For a more extensive introduction to probabilistic logic programming, we refer to De Raedt and Kimmig [2015] or Riguzzi [2023]. We adopt probabilistic Datalog, which is a subset of more general probabilistic programming languages such as ProbLog [Fierens et al., 2015] or dPASP [Geh et al., 2024]. Although our technical presentation focuses on Datalog, it largely carries over to these more general languages. All notation used in the paper is summarized in Table 3 in the Appendix.

**Logic programming.** An *atom* is an expression of the form $r(t_1, \ldots, t_n)$ where $r$ is an $n$-ary *predicate* and each $t_i$ is a *variable* or *constant*. A symbol is a predicate or a constant. A *Datalog rule*, or simply *rule*, is an expression of the form $h \leftarrow b_1 \land \cdots \land b_n$, where the conclusion $h$ is an atom and the premise $b_1 \land \cdots \land b_n$ is a conjunction of $n$ atoms. A *ground* (or non-ground) atom includes no variable (or no constant) argument. A rule is (non-)ground when each occurring atom is (non-)ground. A ground rule with an empty body is also called a *fact*. A *Datalog program*, or simply *program*, $\mathcal{P}$ is a tuple $(\mathcal{F}, \mathcal{R})$, where $\mathcal{F}$ is a set of facts and $\mathcal{R}$ is a set of non-ground rules. The *Herbrand universe* $U$ of $\mathcal{P}$ is the set of all constants that occur in $\mathcal{F}$. The *Herbrand base* $B$ of $\mathcal{P}$ is the set of all ground atoms constructed using the relations and constants in $\mathcal{P}$. Throughout the text, we fix a (probabilistic) logic program $\mathcal{P}$. All formal statements that will follow are w.r.t. this program.

**Example 2.** *Consider the program $\mathcal{P}_{ex}$ including the facts $\mathcal{F}_{ex} = \{r(a, b), r(b, c)\}$ and the rules $\mathcal{R}_{ex} = \{r(X, Y) \leftarrow r(X, Z) \land r(Z, Y)\}$). The Herbrand universe of $\mathcal{P}_{ex}$ is $U = \{a, b, c\}$ and its Herbrand base is $B = \{r(a, a), r(a, b), r(a, c), r(b, a), r(b, b), r(b, c), r(c, a), r(c, b), r(c, c)\}$.*

A set of ground atoms $I \subset B$ satisfies a ground rule $h \leftarrow b_1, \ldots b_n$ if $h \in I$ holds when each $b_i$ is in $I$. The grounding of a rule in $\mathcal{P}$ is the rule that results after consistently replacing each variable with a constant from $U$. The *least model* $M(\mathcal{P})$ of $\mathcal{P}$ is the smallest superset of $\mathcal{F}$ that satisfies each possible grounding of each rule in $\mathcal{R}$. Every program has a unique least model [Ceri et al., 1989]. In Example 2, we have $M(\mathcal{P}_{ex}) = \mathcal{F}_{ex} \cup \{r(a, c)\}$. The program $\mathcal{P}$ entails an atom $\alpha$, written as $\mathcal{P} \models \alpha$, if $\alpha$ is an element of the least model $M(\mathcal{P})$.

**Probabilistic logic programming.** Probabilistic logic programs extend traditional logic programs by treating each fact $f \in \mathcal{F}$ as an independent Bernoulli random variable that becomes true with probability $P_f(f)$, where $P_f : \mathcal{F} \to [0, 1]$. As such, a probabilistic logic program $\mathcal{P}_{\mathbf{p}} = (\mathcal{F}, \mathcal{R}, P_f)$ induces a distribution over all (non-probabilistic) logic programs $(w, \mathcal{R})$, each formed by taking a subset of facts $w$ from $\mathcal{F}$. Each such subset $w \subseteq \mathcal{F}$ is called a *possible world*. This semantics is known as the possible world semantics [Sato, 1995]. The probability $P(\alpha)$ a fact $\alpha$ is true in $\mathcal{P}_{\mathbf{p}}$ is defined as the probability a possible world $w$ leads to the entailment of $\alpha$:

$$P_{\mathcal{F}}(w) \coloneqq \prod_{f \in w} P_f(f) \prod_{f \in \mathcal{F} \setminus w} (1 - P_f(f)) \qquad \text{and} \qquad P(\alpha) \coloneqq \mathbb{E}_{w \sim P_{\mathcal{F}}}\left[(w, \mathcal{R}) \models \alpha\right]. \quad (1)$$

**Example 3.** *The program from Example 2 has four possible worlds: $\{\}$, $\{r(a, b)\}$, $\{r(b, c)\}$, and $\{r(a, b), r(b, c)\}$. Only the last one derives $r(a, c)$. By setting $P_f = \{r(a, b) \mapsto 0.5, r(b, c) \mapsto 0.1\}$, we have $P(r(a, c)) = P_{\mathcal{F}}(r(a, b)) \cdot P_{\mathcal{F}}(r(b, c)) = 0.05$.*

**Equivalence.** A relation is a set of $n$-ary tuples over a domain of elements. An equivalence relation $e$ over a set $S$ is a binary relation $e \subseteq S \times S$ that is *reflexive*, i.e., $\forall \mathsf{x} \in S : e(\mathsf{x}, \mathsf{x})$, *symmetric*, i.e., $\forall \mathsf{x}, \mathsf{y} \in S : e(\mathsf{x}, \mathsf{y}) \Rightarrow e(\mathsf{y}, \mathsf{x})$, and *transitive*, i.e., $\forall \mathsf{x}, \mathsf{y}, \mathsf{z} : e(\mathsf{x}, \mathsf{y}) \wedge e(\mathsf{y}, \mathsf{z}) \Rightarrow e(\mathsf{x}, \mathsf{z})$. Equivalence relations are isomorphic to set partitions, and, hence, we use them interchangeably. We write $\mathcal{E}_S$ to denote the set of all equivalence relations over the set $S$.

We use equivalence relations both over the Herbrand universe (constants) and the Herbrand base (ground atoms) of a program. Following Maene and De Raedt [2023], we define equivalence relations over the constants of the program and not over the atoms. However, each equivalence relation $e_\mathsf{U}$ over the Herbrand universe $\mathsf{U}$ implies an equivalence relation $e_\mathsf{B}$ over the Herbrand base $\mathsf{B}$. In particular,

$$e_\mathsf{B}(\mathsf{r}(\mathsf{a}_1, \ldots, \mathsf{a}_n), \mathsf{r}(\mathsf{b}_1, \ldots, \mathsf{b}_n)) \text{ holds in } \mathsf{B} \text{ if and only if } \bigwedge_{i=1}^{n} e_\mathsf{U}(\mathsf{a}_i, \mathsf{b}_i) \text{ holds in } \mathsf{U}. \qquad (2)$$

The above definition might seem restrictive as atoms with different predicates can never be equivalent. To support such cases, we can simply take predicates as arguments. For example, $\mathsf{r}_1(\mathsf{x}_1, \mathsf{y}_1)$ and $\mathsf{r}_2(\mathsf{x}_2, \mathsf{y}_2)$ become $\mathsf{t}(\mathsf{r}_1, \mathsf{x}_1, \mathsf{y}_1)$ and $\mathsf{t}(\mathsf{r}_2, \mathsf{x}_2, \mathsf{y}_2)$, where $\mathsf{t}$ is a fresh dummy predicate.

## 3 Equivalence in Logic Programming

Intuitively, equivalence creates exchangeability between the facts in the program, meaning that the semantics of the program should remain the same if we replace a fact with an equivalent one, see also Figure 1 (right). For example, if under an equivalence relation $\mathsf{b}$ is equivalent to $\mathsf{c}$, there should be no difference between the rules $\mathsf{a} \leftarrow \mathsf{b}$ and $\mathsf{a} \leftarrow \mathsf{c}$. We formalize equivalence in logic programs using the notion of *saturation* in Definition 2. Then, we formalize soft unification in Definition 3 and show that soft unification violates the semantics of equivalence in probabilistic logic programming and conclude with the impact on learning and inference in Theorems 2 and 3.

**Definition 1** (Set Saturation). *Consider a set $T$ and an equivalence relation $e \in \mathcal{E}_T$ over $T$. The saturation $S_{|e}$ of a set $S \subseteq T$ subject to $e$ is the smallest superset of $S$ that is a union of partitions of $e$.*

$$S_{|e} := \{ y \mid (x, y) \in e, x \in S \} \qquad (3)$$

When a set is its own saturation, i.e., $S = S_{|e}$, we also say that $S$ is saturated subject to $e$. We next expand this definition of saturation from sets to programs.

**Definition 2** (Program Saturation). *For an equivalence relation $e_\mathsf{U} \in \mathcal{E}_\mathsf{U}$, the program $\mathcal{P}_{|e_\mathsf{U}}$ is the saturation of the program $\mathcal{P}$ subject to $e_\mathsf{U}$ if $\mathsf{M}(\mathcal{P}_{|e_\mathsf{U}}) \cap \mathsf{B}$ is the smallest model of $\mathcal{P}$ that is saturated subject to $e_\mathsf{B}$.*

Based on Definition 2, if a saturated program entails a fact $\alpha$, it also entails the fact $\alpha'$ if the two facts are equivalent. We provide an example of saturation below.

**Example 4.** *Consider the equivalence relation $e_\mathsf{U} = \{ (\mathsf{a}, \mathsf{a}), (\mathsf{b}, \mathsf{b}), (\mathsf{c}, \mathsf{c}), (\mathsf{a}, \mathsf{b}), (\mathsf{b}, \mathsf{a}) \}$ and the program $\mathcal{P}_{ex}$ from Example 2. The program $\mathcal{P}_{ex}$ is not saturated subject to $e_\mathsf{U}$, since $(\mathsf{r}(\mathsf{a}, \mathsf{a}), \mathsf{r}(\mathsf{a}, \mathsf{b})) \in e_\mathsf{B}$ and $\mathcal{P} \models \mathsf{r}(\mathsf{a}, \mathsf{b})$, but $\mathcal{P} \not\models \mathsf{r}(\mathsf{a}, \mathsf{a})$. The saturation of the set $\mathcal{F}_{ex} = \{ \mathsf{r}(\mathsf{a}, \mathsf{b}), \mathsf{r}(\mathsf{b}, \mathsf{c}) \}$ subject to $e_\mathsf{B}$ is $\mathcal{F}_{ex|e_\mathsf{B}} = \mathcal{F}_{ex} \cup \{ \mathsf{r}(\mathsf{a}, \mathsf{a}), \mathsf{r}(\mathsf{b}, \mathsf{b}), \mathsf{r}(\mathsf{b}, \mathsf{a}), \mathsf{r}(\mathsf{a}, \mathsf{c}) \}$.*

A standard way to saturate a program is to explicitly axiomatize the equivalence relation and add the corresponding axioms to the program [Fitting, 1996, Chapter 9]. Specifically, for a given equivalence relation $e_\mathsf{U}$, the axiomatization requires introducing (i) a fresh binary predicate $\overset{e_\mathsf{U}}{\approx}$, (ii) all facts in $\mathsf{F}(e_\mathsf{U}) := \{ \mathsf{a} \overset{e_\mathsf{U}}{\approx} \mathsf{b} \mid (\mathsf{a}, \mathsf{b}) \in e_\mathsf{U} \}$ and (3) a set of congruence rules $\mathsf{C}_{e_\mathsf{U}}(\mathcal{R})$:

$$\mathsf{C}_{e_\mathsf{U}}(\mathcal{R}) := \left\{ \mathsf{p}(\mathsf{X}_1, \ldots, \mathsf{X}_n) \leftarrow \mathsf{p}(\mathsf{X}_1', \ldots, \mathsf{X}_n') \wedge \bigwedge_{i=1}^{n} \mathsf{X}_i \overset{e_\mathsf{U}}{\approx} \mathsf{X}_i' \;\middle|\; \mathsf{p} \text{ is a } n\text{-ary predicate in } \mathcal{P} \right\} \qquad (4)$$

In Example 2, there is one congruence rule for each equivalence relation $e_\mathsf{U}$, namely $\mathsf{C}_{e_\mathsf{U}}(\mathcal{R}_{ex}) = \{ \mathsf{r}(\mathsf{X}, \mathsf{Y}) \leftarrow \mathsf{r}(\mathsf{X}', \mathsf{Y}') \wedge \mathsf{X} \overset{e_\mathsf{U}}{\approx} \mathsf{X}' \wedge \mathsf{Y} \overset{e_\mathsf{U}}{\approx} \mathsf{Y}' \}$. We have the following result.

**Theorem 1.** *For any equivalence relation $e_\mathsf{U} \in \mathcal{E}_\mathsf{U}$, the program $(\mathcal{F} \cup \mathsf{F}(e_\mathsf{U}), \mathcal{R} \cup \mathsf{C}_{e_\mathsf{U}}(\mathcal{R}))$ is a saturation $\mathcal{P}_{|e_\mathsf{U}}$ of the program $\mathcal{P}$ subject to $e_\mathsf{U}$.*

Unlike previous work on question answering over logical theories with equality, we do not axiomatize the symmetry and transitivity properties. This is because the facts in $\mathsf{F}(e_\mathsf{U})$ are already an equivalence relation – in Benedikt et al. [2018], the basic assumption is that the input facts $\mathcal{F}$ include no $\approx$-facts.

### 3.1 Soft Unification

Before formalizing soft unification, we will introduce some key notions. Let $\rho := \mathsf{U} \times \mathsf{U}$ be the equivalence relation where all constants are equivalent. We use $\approx$ as a predicate to denote equivalence under $\rho$ and define the set $\mathsf{F}(\rho) := \{\mathsf{c}_1 \approx \mathsf{c}_2 \mid \mathsf{c}_1, \mathsf{c}_2 \in \mathsf{U}\}$.

We now introduce soft unification using the notions from Section 3. Let $\mathsf{C}_\rho(\mathcal{R})$ be the set of rules defined as in (4); however, these rules are defined based on $\approx$. By associating a Bernoulli random variable with each fact in $\mathsf{F}(\rho)$, we can capture the semantics of soft unification using a class of probabilistic logic programs referred to as *soft unification programs*.

**Definition 3** (Soft Unification). *Let $d : \mathbb{R}^k \times \mathbb{R}^k \to [0, 1]$ be a function that returns the similarity between the embeddings $\vec{v}_\mathsf{a}, \vec{v}_\mathsf{b} \in \mathbb{R}^k$ of two symbols $\mathsf{a}$ and $\mathsf{b}$ in $\mathsf{U}$. A soft unification program $\mathcal{P}_\mathbf{s}$ over a set of rules $\mathcal{R}$, a set of facts $\mathcal{F}$, and the similarity function $d$ is defined as the tuple:*

$$\mathcal{P}_\mathbf{s} := (\mathcal{F} \cup \mathsf{F}(\rho), \mathcal{R} \cup \mathsf{C}_\rho(\mathcal{R}), P_d) \tag{5}$$

*The probability a fact $\mathsf{a} \approx \mathsf{b} \in \mathsf{F}(\rho)$ is true in $\mathcal{P}_\mathbf{s}$ is defined as $P_d(\mathsf{a} \approx \mathsf{b}) := d(\vec{v}_\mathsf{a}, \vec{v}_\mathsf{b})$. The probability that a fact in $\mathcal{F}$ is true in $\mathcal{P}_\mathbf{s}$ is simply one[2].*

According to Definition 3, each possible world of a soft unification program is a set of $\approx$-facts, where the probability of each such fact is computed using the constant embeddings. We give an example:

**Example 5.** *The soft unification program given the rules and the facts in the running example includes the facts $\{\mathsf{r}(\mathsf{a},\mathsf{b}), \mathsf{r}(\mathsf{b},\mathsf{c}), \mathsf{a} \approx \mathsf{a}, \mathsf{a} \approx \mathsf{b}, \mathsf{a} \approx \mathsf{c}, \mathsf{b} \approx \mathsf{a}, \mathsf{b} \approx \mathsf{b}, \mathsf{b} \approx \mathsf{c}, \mathsf{c} \approx \mathsf{a}, \mathsf{c} \approx \mathsf{b}, \mathsf{c} \approx \mathsf{c}\}$ and the rules $\{\mathsf{r}(\mathsf{X},\mathsf{Y}) \leftarrow \mathsf{r}(\mathsf{X},\mathsf{Z}) \wedge \mathsf{r}(\mathsf{Z},\mathsf{Y}), \ \mathsf{r}(\mathsf{X},\mathsf{Y}) \leftarrow \mathsf{r}(\mathsf{X}',\mathsf{Y}') \wedge \mathsf{X} \approx \mathsf{X}' \wedge \mathsf{Y} \approx \mathsf{Y}'\}$. One possible world of this program is $\{\mathsf{a} \approx \mathsf{a}, \mathsf{a} \approx \mathsf{b}, \mathsf{b} \approx \mathsf{c}\}$.*

In the appendix, we discuss how Definition 3 can represent different soft unification techniques [Rocktäschel and Riedel, 2017, Minervini et al., 2020a, Maene and De Raedt, 2023].

*Do soft unification programs adhere to the properties of equivalence relations?* From Definition 3, it follows that to enforce a soft unification program to treat equivalence as a reflexive relation, it suffices to choose the similarity function $d$ such that $d(\vec{v}_\mathsf{c}, \vec{v}_\mathsf{c}) = 1$ for every embedded symbol $\vec{v}_\mathsf{c}$. Symmetry can similarly be enforced, e.g. by defining $\approx$ as an unordered predicate. However, soft unification programs do not treat equivalence as a transitive relation, failing to capture the true semantics of equivalence. Returning to Example 5, there are possible worlds in which the facts $\mathsf{a} \approx \mathsf{b}$ and $\mathsf{b} \approx \mathsf{c}$ hold, but the fact $\mathsf{a} \approx \mathsf{c}$ does not hold. Nonetheless, under a mild assumption on the similarity function $d$, we get a weak kind of transitivity due to the use of embeddings.

**Theorem 2.** *For each possible world $w \subseteq \mathsf{F}(\rho)$ of a soft unification program $\mathcal{P}_\mathbf{s}$ and each similarity function $d$ satisfying*

$$d(\vec{v}_1, \vec{v}_2) = 1 \text{ if and only if } \vec{v}_1 = \vec{v}_2, \forall \vec{v}_1, \vec{v}_2 \in \mathbb{R}^k, \tag{6}$$

*$P(w) = 1$ implies that the $\approx$-facts in $w$ satisfy the semantics of transitivity.*

Theorem 2 implies that a soft unification program distribution can only place arbitrarily high probability mass on possible worlds where the $\approx$-facts satisfy the semantics of transitivity. For example, there exists no embedding of the constants $\mathsf{a}$, $\mathsf{b}$, and $\mathsf{c}$ that satisfies the following probabilities: $P(\mathsf{a} \approx \mathsf{b}) = 1$, $P(\mathsf{b} \approx \mathsf{c}) = 1$, and $P(\mathsf{a} \approx \mathsf{c}) = 0$. From a Bayesian viewpoint, the probability mass on the possible worlds whose $\approx$-facts do not satisfy the semantics of transitivity is essentially "leaked". Indeed, after training convergences, these possible worlds need to have probability zero, see also Van Krieken et al. [2024].

Soft unification also poses challenges in learning. It is known that any distribution over Boolean variables is a multilinear polynomial function [Darwiche, 2003]. Hence, all optima are global, making it straightforward to learn neurosymbolic models computing the gradients of those functions, such as DeepProbLog [Manhaeve et al., 2021]. The above does not hold when using soft unification.

**Theorem 3.** *The probability $P(\alpha)$ a ground atom $\alpha \notin \mathcal{F}$ is true in a soft unification program $\mathcal{P}_\mathbf{s}$ is not a multilinear polynomial function in the embeddings.*

---

[2]Conventionally, the facts in a soft unification program are non-probabilistic. However, it is also possible to associate a probability to them as in probabilistic logic programming, see (1).

# 4 Semantics of Equivalence in Probabilistic Logic Programming

This section introduces our semantics, referred to as *probabilistic equivalence semantics*. Our semantics relies on a distribution over the equivalence relations $P_{\mathcal{E}} : \mathcal{E}_{\mathsf{U}} \to [0, 1]$. Recall that possible world semantics relies on a distribution over the possible worlds $P_{\mathcal{F}} : 2^{\mathcal{F}} \to [0, 1]$, Section 2. As in soft unification, we use $\approx$ as a predicate to denote equivalence in $\rho$ and assume no $\approx$-atom occurs in $\mathcal{R}$ or $\mathcal{F}$.

**Definition 4** (Probabilistic Equivalence Semantics). *A probabilistic equivalence program $\mathcal{P}_{\mathbf{e}}$ over a set of rules $\mathcal{R}$, a set of facts $\mathcal{F}$, and a distribution $P_{\mathcal{E}} : \mathcal{E}_{\mathsf{U}} \to [0, 1]$ is defined as the tuple $(\mathcal{R}, \mathcal{F}, P_{\mathcal{E}})$. The probability a fact $\mathsf{a} \approx \mathsf{b} \in \mathsf{F}(\rho)$ is true in $\mathcal{P}_{\mathbf{e}}$ is defined as the probability $(\mathsf{a}, \mathsf{b})$ belongs to some $e_{\mathsf{U}} \in \mathcal{E}_{\mathsf{U}}$:*

$$P(\mathsf{a} \approx \mathsf{b}) := \mathbb{E}_{e_{\mathsf{U}} \sim P_{\mathcal{E}}} [(\mathsf{a}, \mathsf{b}) \in e_{\mathsf{U}}] \tag{7}$$

*The probability a non-$\approx$ fact $\alpha$ is true in $\mathcal{P}_{\mathbf{e}}$ is defined as the probability $\alpha$ is entailed by a saturation $\mathcal{P}_{|e_{\mathsf{U}}}$ of $(\mathcal{R}, \mathcal{F})$ subject to some $e_{\mathsf{U}} \in \mathcal{E}_{\mathsf{U}}$ :*

$$P(\alpha) := \mathbb{E}_{e_{\mathsf{U}} \sim P_{\mathcal{E}}} \left[ \mathcal{P}_{|e_{\mathsf{U}}} \models \alpha \right] \tag{8}$$

Section 4.1 provides a definition of $P_{\mathcal{E}}$ based on an embedding model of constants. We provide an example of Definition 4 below.

**Example 6.** *Consider the running example with a distribution over the five possible equivalence relations on $\mathsf{U} = \{\mathsf{a}, \mathsf{b}, \mathsf{c}\}$.*

$$P_{\mathcal{E}} = \left\{ \boxed{\text{(a)(b) (c)}} \mapsto 0.4, \ \boxed{\text{(a\_b) (c)}} \mapsto 0.3, \ \boxed{\text{(a)(b) c}} \mapsto 0.0, \ \boxed{\text{(a)(b) c}} \mapsto 0.2, \ \boxed{\text{a b c}} \mapsto 0.1 \right\}$$

*The probability of $\mathsf{r}(\mathsf{b}, \mathsf{a})$ under Definition 4 is $P(\mathsf{r}(\mathsf{b}, \mathsf{a})) = 0.3 + 0.0 + 0.1 = 0.4$, since either $\mathsf{a} \approx \mathsf{b}$ or $\mathsf{a} \approx \mathsf{c}$ must hold to derive $\mathsf{r}(\mathsf{b}, \mathsf{a})$ from the least model $\mathsf{M}(\mathcal{P}_{ex}) = \{\mathsf{r}(\mathsf{a}, \mathsf{b}), \mathsf{r}(\mathsf{b}, \mathsf{c}), \mathsf{r}(\mathsf{a}, \mathsf{c})\}$.*

The probabilistic equivalence semantics (Definition 4) can be seen as the possible world semantics (Definition 1) by defining $P(w) := \mathbb{E}_{e_{\mathsf{U}} \sim P_{\mathcal{E}}}[w = \mathcal{F}_{|e_{\mathsf{B}}}]$. However, unlike in the possible world semantics, the facts in each possible $w$ are no longer independent. We further discuss this in Appendix C.

## 4.1 Factorizing the Distribution Over Equivalence Relations

We now discuss how to represent an equivalence distribution $P_{\mathcal{E}} : \mathcal{E}_{\mathsf{U}} \to [0, 1]$ (see Definition 4) over embeddings of constants. Similarly to representing the full joint distribution $P_{\mathcal{F}}$ over all possible worlds, we need to make additional assumptions on $P_{\mathcal{E}}$ to make the representation of $P_{\mathcal{E}}$ tractable, as the size of $\mathcal{E}_{\mathsf{U}}$, known as the Bell number, grows exponentially in the size of $\mathsf{U}$. Inspired by probabilistic clustering [Deng and Han, 2018], we associate an independent latent random variable $V_{\mathsf{c}}$ with each constant $\mathsf{c} \in \mathsf{U}$. Hence, the probability $\mathsf{c}$ belongs to the $i^{\text{th}}$ partition is $P(V_{\mathsf{c}} = i)$. We can treat the probability vector of the categorical $V_{\mathsf{c}}$ as an embedding for the constant $\mathsf{c}$.

$$P_{\mathcal{E}}(e_{\mathsf{U}}) := \sum_{\pi_e} \prod_{\mathsf{c} \in \mathsf{U}} P(V_{\mathsf{c}} = \pi_e(\mathsf{c})) \tag{9}$$

Here, $\pi_e : \mathsf{U} \to \{1, \dots, k\}$ is any function such that $\forall (\mathsf{a}, \mathsf{b}) \in e_{\mathsf{U}} : \pi_e(\mathsf{a}) = \pi_e(\mathsf{b})$, with $k$ being the embedding dimension. Although $\pi_e$ is required for the above definition of $P_{\mathcal{E}}(e_{\mathsf{U}})$ to marginalize away the possible orderings of the partitions, it never needs to be computed in practice. An advantage of this factorization is that we can trivially calculate the expectation of two atoms being equivalent.

**Corollary 1.** *When $P_{\mathcal{E}}$ is defined as in (9), the probability an equivalence atom $\mathsf{a} \approx \mathsf{b}$ is true in a probabilistic equivalence program $\mathcal{P}_{\mathbf{e}} = (\mathcal{R}, \mathcal{F}, P_{\mathcal{E}})$ is given by:*

$$P(\mathsf{a} \approx \mathsf{b}) = \sum_{e_{\mathsf{U}} \in \mathcal{E}_{\mathsf{U}}} \mathbb{1}[(\mathsf{a}, \mathsf{b}) \in e_{\mathsf{U}}] \sum_{\pi_e} \prod_{\mathsf{c} \in \mathsf{U}} P(V_{\mathsf{c}} = \pi_e(\mathsf{c})) = \sum_{i \in \{1, \dots, k\}} P(V_{\mathsf{a}} = i) P(V_{\mathsf{b}} = i)$$

Our factorization assumes that the probability a constant is part of a partition is independent, reducing the expressivity. Specifically, we prove below that it can no longer represent any combination of marginal probabilities between constants.

**Theorem 4.** *Consider a matrix $M$ that contains all the marginal probabilities of constants being equivalent under the probabilistic equivalence semantics, i.e., $M_{i,j} = P(c_i \approx c_j)$ with $c_i, c_j \in U$. If $P_{\mathcal{E}}$ is factorized as in (9), $M$ is positive semi-definite.*

According to Theorem 4, the marginals of equivalence facts should always form a positive semi-definite matrix $M$, or they cannot be expressed by our factorization (9).

## 5 Inference & Learning

This section discusses exact and approximate techniques for inference using probabilistic equivalence programs over a constant embedding space. We start by briefly discussing probabilistic Datalog inference. Then, we discuss the changes required to support our semantics.

**Probabilistic Datalog.** Datalog inference is typically based on the fixed-point semantics [Ceri et al., 1989]. Starting with a given set of facts $\mathcal{F}$, at each inference step, we apply the rules in $\mathcal{R}$ to the existing facts until no new facts can be derived. Modern Datalog engines rely on techniques such as semi-naive evaluation [Bancilhon, 1986] and Trigger Graphs [Tsamoura et al., 2021] to reduce the number of rule applications deriving no new facts. By keeping track of the rules applied to derive a targetfact, we can compute the *lineage* of the fact, i.e., the Boolean formula representing the possible worlds that lead to its derivation. Calculating the probability of the target fact then reduces to calculating the probability of this Boolean formula [Fierens et al., 2015, Vlasselaer et al., 2016], a problem known as *weighted model counting* [Sang et al., 2005].

**Singularization.** Consider a single equivalence relation. The naive approach to reason under this relation is to introduce a special predicate $\approx$ and add the congruence rules $C(\mathcal{R})$ in (4) to the program, Theorem 1. More advanced techniques deal with equivalence by taking advantage of the symmetry between elements in the same partition [Fitting, 1996, Chapter 9]. This idea was implemented as *paramodulation* [Robinson and Wos, 1983] and *singularization* [Marnette, 2009]. We briefly explain the latter technique below, as it is most suited to Datalog.

**Definition 5** (Singularization [Marnette, 2009]). *The singularization $Sg(\mathcal{R}, p)$ of a set of rules $\mathcal{R}$ with respect to a target predicate $p$ is the set that includes each rule $r \in \mathcal{R}$ transformed as follows: for each variable $X$ that occurs more than once in an non-$\approx$-atom in the premise of $r$, $X$ is replaced with a unique variable $X_i$ and the atom $X \approx X_i$ is added to the premise of $r$. In addition, $Sg(\mathcal{R}, p)$ includes the congruence rule $p(X_1', \ldots, X_n') \leftarrow p(X_1, \ldots, X_n) \wedge X_1 \approx X_1' \wedge \cdots \wedge X_n \approx X_n'$.*

In Definition 5, the predicate $p$ is different from $\approx$, it is defined by exactly one rule and cannot occur in the body of any rule in $\mathcal{R}$. This assumption is without loss of generality, as it can always be enforced by introducing fresh predicates. In our running example, singularization turns the transitive rule $r(X, Y) \leftarrow r(X, Z) \wedge r(Z, Y)$ into $r(X, Y) \leftarrow r(X, Z_1) \wedge r(Z_2, Y) \wedge Z_1 \approx Z_2$. Singularization alters the least model and does not preserve probability in general. However, it leads to sound inference under our semantics (see Theorem 5 below).

**Magic sets transformation.** A further well-known optimization in Datalog inference is goal-driven inference. Throughout the rest of this section, consider a target predicate $p$ as in Definition 5 and a ground atom $p(\mathbf{c})$, where $\mathbf{c}$ is a tuple of constants. The goal is to find whether the program entails $p(\mathbf{c})$. To do that, instead of computing the least model entirely, specific atoms, called *magic atoms*, prevent the application of a rule until it is necessary for the derivation of $p(\mathbf{c})$. This approach is known as the *magic sets transform* [Beeri and Ramakrishnan, 1987], which we denote as $Mag(\mathcal{R}, p(\mathbf{c}))$. Appendix D demonstrates the magic transform in the running example. As we have a regular Datalog program after singularization, we can apply magic sets out of the box, bringing us to the final program.

**Theorem 5.** *For each distribution $P_{\mathcal{E}} : \mathcal{E}_U \to [0, 1]$ and each target ground $p$-atom $\alpha$, we have*

$$\mathbb{E}_{e_U \sim P_{\mathcal{E}}} \left[ \mathcal{P}_{|e_U} \models \alpha \right] = \mathbb{E}_{e_U \sim P_{\mathcal{E}}} [(\mathcal{F} \cup F(e_U), Mag(Sg(\mathcal{R}, p), \alpha)) \models \alpha].$$

The exact inference procedure is summarized in Algorithm 1. We assume that distributions $P_{\mathcal{E}}$ are represented over constant embeddings as in Section 4.1. The function $Lin$ computes the lineage of the target. The lineage of the Datalog program contains only $\approx$-facts, due to the latent variable representation (see Section 4.1). These $\approx$-facts are encoded into a Boolean formula whose variables have probabilities encoded in a vector $\mathbf{W}$ using the function $BooleanEnc$, see Cao et al. [2023], De Smet and Zuidberg Dos Martires [2024]. Finally, $WMC$ computes the weighted model counting of the resulting formula subject to the weights.

| **Algorithm 1:** Exact Inference | **Algorithm 2:** Approximate Inference |
|---|---|
| **Input:** Program $\mathcal{P} = (\mathcal{F}, \mathcal{R})$, equivalence distribution $P_{\mathcal{E}}$ as in (9), and target p-fact $\alpha$. | **Input:** Program $\mathcal{P} = (\mathcal{F}, \mathcal{R})$, equivalence distribution $P_{\mathcal{E}}$ as in (9), target p-fact $\alpha$, and number of samples $k$. |
| **Output:** Probability $P(\alpha)$ | **Output:** Approximation of $P(\alpha)$. |
| **Step 1: Singularization** 
 $\quad \mathcal{R}' := \mathsf{Sg}(\mathcal{R}, \mathsf{p})$ 
 **Step 2: Magic Sets** 
 $\quad \mathcal{R}'' := \mathsf{Mag}(\mathcal{R}', \alpha)$ 
 **Step 3: Lineage Computation** 
 $\quad \mathsf{lineage} := \mathsf{Lin}(\mathcal{F} \cup \mathsf{F}(\mathsf{U} \times \mathsf{U}), \mathcal{R}'', \alpha)$ 
 **Step 4: Boolean Encoding** 
 $\quad \phi_\alpha, \mathbf{W} := \mathsf{BooleanEnc}(\mathsf{lineage}, \mathcal{E}_\mathsf{U})$ 
 **Step 5: Weighted Model Counting** 
 $\quad P(\alpha) := \mathsf{WMC}(\phi_\alpha, \mathbf{W})$ | **Step 1: Magic Sets** 
 $\quad \mathcal{R}' := \mathsf{Mag}(\mathcal{R}, \alpha)$ 
 **Step 2: Sampling the Latents** 
 $\quad \mathbf{V} \sim P_{\mathcal{E}}$ 
 **Step 3: Constant Abstraction** 
 $\quad \mathcal{F}' := \mathsf{Inst}(\mathcal{F}, \mathbf{V})$ 
 **Step 4: Forward Inference** 
 $\quad x := \mathsf{CheckEntails}(\mathcal{F}', \mathcal{R}', \mathsf{Inst}(\alpha, \mathbf{V}))$ 
 **Step 5: Repeat** steps 2–4, collecting $k$ 
 $\quad$ samples $[x_1, x_2, \ldots, x_k]$. |
| **return** $P(\alpha)$ | **return** $\mathsf{mean}([x_1, x_2, \ldots, x_k])$ |

**Approximate Inference.** As exact probabilistic inference is often intractable, we also include a straightforward Monte-Carlo approximation, similar to Gutmann et al. [2011]. In our case, the probability of a ground atom $\alpha$ is estimated as the average number of times the $\alpha$ is entailed (i.e., $\mathcal{P}_{|e_\mathsf{U}} \models \alpha$) under the sampled equivalence relations $e_\mathsf{U}$. In particular, let $\mathsf{Inst}(\mathcal{F}, \mathbf{V})$ be the set of facts that results after replacing each constant $\mathsf{c} \in \mathsf{U}$ in each fact in $\mathcal{F}$ with the value of its latent $V_\mathsf{c}$ in $\mathbf{V}$. Here, we write $\mathbf{V}$ for an assignment to all the latent variables of the constants in $\mathsf{U}$. After applying the instantiation $\mathsf{Inst}$, we obtain a regular Datalog program. So applying singularization is no longer necessary.

In Lemma 4, we prove that such an instantiation is a valid saturation of the program. Algorithm 2 contains the pseudo-code and we prove the soundness of our sampling technique below.

**Theorem 6.** *For each distribution* $P_{\mathcal{E}} : \mathcal{E}_\mathsf{U} \to [0, 1]$ *and each target ground* p-*atom* $\alpha$, *we have*

$$\mathbb{E}_{e_\mathsf{U} \sim P_{\mathcal{E}}} \left[ \mathcal{P}_{|e_\mathsf{U}} \models \alpha \right] = \lim_{k \to \infty} \frac{1}{k} \sum_{i=1}^{k} \mathbb{1}[(\mathsf{Inst}(\mathcal{F}, \mathbf{V}_i), \mathsf{Mag}(\mathcal{R}, \alpha)) \models \mathsf{Inst}(\alpha, \mathbf{V}_i)], \text{ where } \mathbf{V}_i \sim P_{\mathcal{E}}.$$
(10)

**Learning.** Using exact inference, the computation for $P(\alpha)$ is end-to-end differentiable, and we can hence learn the embeddings from entailment using gradient descent on the negative log-likelihood $(-\log P(\alpha))$. When sampling for approximate inference, we lose differentiability and require a gradient estimator. Our experiments use the score function estimator (also known as REINFORCE) in combination with the leave-one-out baseline for variance reduction [Kool et al., 2019].

## 6 Experiments

We assess the effect of our semantics on two settings where embeddings and rules need to be learnt jointly. First, we consider link prediction in knowledge graphs. Second, we consider differentiable finite state machines, where the structure of a finite state machine needs to be learned from sequences of subsymbolic data.

### 6.1 Link Prediction in Knowledge Graphs

**Benchmarks.** We use two well-known small knowledge graphs: countries [Bouchard et al., 2015] and nations [Rummel, 1992]. Countries contains the locations of countries and regions in the world and comes in three variants with increasing difficulty (S1, S2, and S3). Nations contains geopolitical relations between countries. There are no standard data splits for nations, which complicates the comparison with prior results. We adopt the same splits as Rocktäschel and Riedel [2017] and Minervini et al. [2020b].

Table 1: Link prediction results on the countries and nations knowledge graphs. We report the mean and standard deviation over 10 seeds. Baselines are taken from Minervini et al. [2020b,a], Maene and De Raedt [2023]. Tables 5 and 6 in the appendix show the used hyperparameters.

| Dataset | | Metric | NTP | GNTP | CTP | DeepSoftLog | Ours |
|---|---|---|---|---|---|---|---|
| | S1 | AUC-PR | $90.83 \pm 15.4$ | $99.98 \pm 0.05$ | $\mathbf{100.0} \pm 0.00$ | $\mathbf{100.0} \pm 0.00$ | $\mathbf{100.0} \pm 0.00$ |
| Countries | S2 | AUC-PR | $87.40 \pm 11.7$ | $90.82 \pm 0.88$ | $91.81 \pm 1.07$ | $97.67 \pm 0.98$ | $\mathbf{100.0} \pm 0.00$ |
| | S3 | AUC-PR | $56.68 \pm 17.6$ | $87.70 \pm 4.79$ | $94.78 \pm 0.00$ | $97.90 \pm 1.00$ | $\mathbf{99.89} \pm 0.31$ |
| | | MRR | 0.61 | 0.658 | $0.709 \pm 0.03$ | timeout | $\mathbf{0.724} \pm 0.01$ |
| Nations | | Hits@1 | 0.45 | 0.493 | $0.562 \pm 0.05$ | timeout | $\mathbf{0.591} \pm 0.02$ |
| | | Hits@3 | 0.73 | 0.781 | $0.813 \pm 0.03$ | timeout | $\mathbf{0.819} \pm 0.01$ |
| | | Hits@10 | 0.87 | 0.985 | $\mathbf{0.995} \pm 0.00$ | timeout | $0.988 \pm 0.01$ |

**Evaluation.** As is standard in the literature, we report the area under the precision-recall curve (AUC-PR) for the countries knowledge graph and use filtered ranking [Bordes et al., 2013] for nations. For each fact $p(a, b)$ in the test dataset, we take all possible corrupted facts by replacing one of the two arguments, creating $p(a', b)$ and $p(a, b')$, and filter out any corrupted facts that appear in the knowledge graph. We then rank the probability of the test fact compared to all these corrupted facts. When different facts have the same probability, previous works often rank ties in their favor. This is known to skew results [Sun et al., 2020], and we instead do random tie-breaking. Using the ranks, we report the mean reciprocal rank and hits@$k$ (with $k \in \{1, 3, 10\}$).

**Setup.** We train with the same rule templates as prior work (c.f. Appendix B) and set the embedding dimension to the vocabulary size to avoid limiting the expressivity. We use approximate inference as described in Section 5 with the GLog Datalog engine [Tsamoura et al., 2021]. The only biased aspect of optimization is that we truncate the fixed-point iteration of the Datalog engine, meaning the derivation of each fact can take at most 8 steps. This was found to speed up training with no significant performance penalty. We train for 2 epochs with the Adam optimizer. All hyperparameters are summarized in Appendix B. As baselines, we compare with prior work that reasons on embeddings in logic programs. Namely, the Neural Theorem Prover (NTP) [Rocktäschel and Riedel, 2017], the Greedy Neural Theorem Prover (GNTP) [Minervini et al., 2020b], the Conditional Theorem Prover (CTP) [Minervini et al., 2020b], and DeepSoftLog [Maene and De Raedt, 2023].

**Results.** The results are summarized in Table 1. On every tested knowledge graph, we outperform existing soft unification models. Notably, this is still the case for CTP, which is not restricted to a specific set of rule templates, as all other tested methods are.

### 6.2 Differentiable Finite State Machines

Differentiable finite state machines classify an input sequence of MNIST images into accepting and non-accepting sequences. For example, in the language $(01)^*$,  and  are accepting sequences while  and  are non-accepting sequences. The goal is to jointly train a neural network, in this case a digit classifier, and the transition rules of the state machine.

We follow the same experimental setting as Maene and De Raedt [2023], using three different binary languages: the language of 01 repetitions, the language of sequences with exactly one 1, and the language of sequences with an even number of 1's. In each case, 20 example digits are given as concept supervision to provide a non-random baseline accuracy to the neural network. After this, the models are trained on sequences of length 4. To test generalization, the test sequences have double the length and use images disjoint from the training split. Table 2 displays the results. As a baseline, we include both DeepSoftLog and a simple RNN. On each language, the introduction of transitivity greatly improves the results of DeepSoftLog, to the point of saturating this benchmark.

## 7 Related Work

**Equivalence & logic.** Equivalence in logic has been studied extensively [Nilsson and Maluszynski, 1995, Chap. 13; Fitting, 1996, Chap. 9], going back to Jaffar et al. [1984]. Logic with equivalence relations has been used in the context of database dependencies [Marnette, 2009, Benedikt et al., 2018], the semantic web [Motik et al., 2015], compiler optimizations [Zhang et al., 2023], or second-

Table 2: AUC-PR on sequence classification with differentiable automata for three grammars. We report the mean and standard deviation over 10 seeds. Baselines are from Maene and De Raedt [2023]

| Language | (01)* | 0*10* | (0 | 10*1)* |
|---|---|---|---|
| RNN | $77.63 \pm 15.05$ | $61.59 \pm 10.09$ | $50.14 \pm 1.36$ |
| DeepSoftLog | $83.93 \pm 25.87$ | $87.01 \pm 7.18$ | $56.12 \pm 15.98$ |
| Ours | $\mathbf{99.60} \pm 1.02$ | $\mathbf{97.46} \pm 5.87$ | $\mathbf{99.61} \pm 0.54$ |

order theories [Tsamoura and Motik, 2024]. To the best of our knowledge, we are the first to place *uncertainty* on the equivalence. Exchangeability in probability distributions was also studied under the name of lifted inference [Niepert and Van den Broeck, 2014]. In lifted inference, there is exchangeability between the variables of the probability distribution, while in our work, we have a probability distribution on the exchangeability between symbols.

**Soft unification.** Many extensions of logic have been proposed to handle uncertainty or vagueness. Similarity-based logics extend fuzzy logic with a similarity on symbols [Gerla and Sessa, 1999, Fontana and Formato, 2002, Godo and Rodríguez, 2008]. This fuzzy matching between symbols has been variously called soft or weak unification, as it can be seen as a relaxation of the unification algorithm that matches symbols during SLD-resolution. A key difference between the similarity used in soft unification and the equivalence we consider is the former does not respect transitivity. Further works have studied the model-based semantics of soft unification [Sessa, 2002, Medina et al., 2004] and the use of similarities or proximities [Medina et al., 2004, Julián-Iranzo and Rubio-Manzano, 2015] and introduced practical implementations such as Bousi~Prolog [Julián-Iranzo et al., 2009].

Related to soft unification, Cohen [2000a,b] relaxed database joins using similarity. Rocktäschel and Riedel [2017] took a neurosymbolic view on soft unification with the Neural Theorem Prover (NTP) by learning the embeddings in an end-to-end differentiable way. Minervini et al. [2020a] improved the scalability of the NTP using greedy approximations. Maene and De Raedt [2023] argued for using probabilistic logic instead of the fuzzy semantics, as the fuzzy inference leads to poor optimization [de Jong and Sha, 2019]. All these works still rely on similarity and do not use full equivalence. Lastly, relational graph neural network architectures may approximate the behaviour of NTPs through message passing [Barbiero et al., 2024].

**Knowledge graph embeddings.** Embeddings are the standard approach in knowledge graphs to handle incompleteness [Bordes et al., 2013]. From a probabilistic view, embeddings can be seen as a low-rank decomposition of the distribution over triples [Loconte et al., 2023]. In contrast to our work, the probability of an atom here only relies on its own embeddings (up to normalization). Moreover, these probabilistic semantics of embeddings do not allow rule learning. Several recent knowledge graph methods have proposed neurosymbolic rule learning [DeLong et al., 2024]. Unlike our work, these are typically not end-to-end differentiable and do not have sound probabilistic semantics.

## 8 Conclusion & Limitations

We investigated the semantics of soft unification for learning and reasoning over embeddings. We showed that soft unification does not adhere to the properties of logical equivalence causing issues in inference and learning. To overcome those issues, we introduced the equivalence semantics in logic programming. We linked our semantics to reasoning over constant embeddings by associating each constant with a category corresponding to the partition the constant belongs. We also adapted existing techniques for inference in the presence of equality to reason under our semantics. Lastly, we experimentally confirmed that enforcing transitivity can considerably improve performance on neurosymbolic tasks, where the rules and embeddings are learned jointly.

Several open questions and limitations remain. First, developing lower variance methods for gradient estimation is necessary for effective training with a large number of rules. Full grounding is often infeasible when learning rules, so many existing approximation strategies are not applicable. Second, our approach still relies on templates for rule learning, so either considerable language bias is incurred or inference becomes challenging. Third, the extension to other languages, such as non-monotonic logic programming or declarative probabilistic languages, is left to future work.

## Acknowledgements

Jaron Maene received funding from the Flemish Government (AI Research Program). We thank Luc De Raedt for his valuable feedback.

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

Table 3: The notation used in our paper.

| Symbol | Ref. | Description |
| --- | --- | --- |
| $c \in U$ | Sec. 2 | A constant in $U$. |
| $U$ | Sec. 2 | The Herbrand universe of $\mathcal{P}$. |
| $X$ | Sec. 2 | A variable ranging over the elements of $U$. |
| $\alpha \in B$ | Sec. 2 | A ground atom in $B$. |
| $B$ | Sec. 2 | The Herbrand base of $\mathcal{P}$. |
| $f \in \mathcal{F}$ | Sec. 2 | A fact in $\mathcal{F}$. |
| $\mathcal{F}$ | Sec. 2 | Set of facts, i.e., ground atoms. |
| $h \leftarrow b_1, \ldots, b_n$ | Sec. 2 | Datalog rule. |
| $\mathcal{R}$ | Sec. 2 | Set of non-ground rules. |
| $\mathcal{P} := (\mathcal{R}, \mathcal{F})$ | Sec. 2 | Datalog program. |
| $\mathcal{P}_{ex}$ | Ex. 2 | Program in the running example. |
| $M(\mathcal{P})$ | Sec. 2 | Least Herbrand model of $\mathcal{P}$. |
| $\mathcal{P} \models \alpha$ | Sec. 2 | Entailment, ground atom $\alpha$ is entailed by $\mathcal{P}$. |
| $\mathcal{E}_U$ | Sec. 2 | The set of all equivalence relations over the set $U$. |
| $e_U$ | Sec. 3 | An equivalence relation over $U$. |
| $e_B$ | Sec. 3 | An equivalence relation over $B$. |
| $\overset{e_U}{\approx}$ | Sec. 3 | Predicate to denote equivalence under $e_U$. |
| $\rho := U \times U$ | Sec. 3.1 | The equivalence relation in which all elements are equivalent. |
| $\approx$ | Sec. 3.1 | Predicate to denote equivalence under $\rho$. |
| $S_{|e_U}$ | Def. 1 | Saturation of the set $S$ subject to the equivalence relation $e_U$. |
| $\mathcal{P}_{|e_U}$ | Def. 2 | Saturation of the program $\mathcal{P}$ subject to the equivalence relation $e_U$. |
| $P_f : \mathcal{F} \to [0,1]$ | Sec. 2 | Probability for each fact in $\mathcal{F}$. |
| $P_d : \mathsf{F}(\rho) \to [0,1]$ | Def. 2 | Probability for each equality fact in $\mathsf{F}(\rho)$. |
| $\mathcal{P}_{\mathbf{p}} := (\mathcal{F}, \mathcal{R}, P_f)$ | Sec. 2 | Probabilistic logic program. |
| $w \subseteq \mathcal{F}$ | Sec. 2 | Possible world, subset of the facts $\mathcal{F}$. |
| $2^{\mathcal{F}}$ | Sec. 2 | Set of all possible subsets of $\mathcal{F}$. |
| $P_{\mathcal{F}} : 2^{\mathcal{F}} \to [0,1]$ | Eq. 1 | Probability distribution over all possible worlds of the probabilistic logic program $\mathcal{P}$. |
| $P_{\mathcal{E}} : \mathcal{E}_U \to [0,1]$ | Eq. 9 | Probability distribution over $\mathcal{E}_U$. |
| $\mathsf{F}(e_U) := \{a \overset{e_U}{\approx} b \mid (a,b) \in e_U\}$ | Sec. 3 | Set of equivalence facts for $e_U$. |
| $\mathsf{C}_{e_U}(\mathcal{R})$ | Eq. 4 | Set of congruence rules subject to $e_U$. |
| $\mathcal{P}_{\mathbf{s}} := (\mathcal{F} \cup \mathsf{F}(\rho), \mathcal{R} \cup \mathsf{C}_\rho(\mathcal{R}), P_d)$ | Def. 3 | Soft unification program. |
| $\mathcal{P}_{\mathbf{e}} := (\mathcal{R}, \mathcal{F}, P_{\mathcal{E}})$ | Def. 4 | Probabilistic equivalence program. |
| $V_c$ | Sec. 4.1 | Latent categorical variable for the constant $c \in U$. |
| $\mathsf{Sg}(\mathcal{R}, p)$ | Def. 5 | Singularization of the rules in $\mathcal{R}$ with respect to a target predicate $p$. |
| $\mathsf{Mag}(\mathcal{R}, \alpha)$ | Sec. 5 | Magic sets transformation of the rules in $\mathcal{R}$ subject to the fact $\alpha$. |
| $\mathbf{V}$ | Sec. 5 | Assignment to all latent variables $V_c$, for $c \in U$. |
| $\mathsf{Inst}(\mathcal{F}, \mathbf{V})$ | Sec. 5 | The set of facts that results after replacing each constant $c \in U$ in each fact in $\mathcal{F}$ with the value of the latent $V_c$ w.r.t. $\mathbf{V}$. |

# A  Proofs

We first establish some preliminary definitions and theorems regarding lattices, fixed points, and monotone logic programming. We then use this background to establish that both models and saturated models of Datalog programs form lattices. These results are later used to prove the main theorems from the paper.

Table 4: The additional notation used in our appendix.

| Symbol | Description |
|---|---|
| $\theta$ | Grounding substitution, i.e., mapping of variables to constants. |
| $a\theta$ | Ground atom that results after replacing each variable $X$ in a with $\theta(X)$. |
| $\mathcal{M}(\mathcal{P})$ | Set of all models for a Datalog program $\mathcal{P}$. |
| $\sqsubseteq$ | Partial order. |
| $(S, \sqsubseteq)$ | Partially order set (poset) for set $S$. |
| $\inf T$ | The greatest lower bound of $(T, \sqsubseteq)$. |
| $T_{\mathcal{P}} : 2^{\mathsf{B}} \to 2^{\mathsf{B}}$ | The immediate consequence operator. |
| $\mathsf{Fix} : (S \to S) \to 2^S$ | Fixed point operator over set $S$. |
| $\mathcal{M}_{e_{\mathsf{U}}}(\mathcal{P})$ | Set of all saturated models of $\mathcal{P}$ subject to $e_{\mathsf{U}}$. |
| $\inf \mathcal{M}_{e_{\mathsf{U}}}(\mathcal{P})$ | Least saturated model of $\mathcal{P}$ subject to $e_{\mathsf{U}}$. |

## A.1 Preliminaries

**Definition 6** (Partial Order). *A partial order over a set $S$ is a binary relation $\sqsubseteq$ that is* reflexive, *i.e., $\forall a \in S : a \sqsubseteq a$,* antisymmetric, *i.e., $\forall a, b \in S : a \sqsubseteq b \wedge b \sqsubseteq a \Rightarrow a = b$, and* transitive, *i.e., $\forall a, b, c \in S : a \sqsubseteq b \wedge b \sqsubseteq c \Rightarrow a \sqsubseteq c$. A partially ordered set or poset is a tuple of the form $(S, \sqsubseteq)$, where $S$ is a set and $\sqsubseteq$ is a partial order.*

For our purposes, we assume all sets are finite.

**Definition 7** (Lower Bound & Infimum). *Let $(S, \sqsubseteq)$ be a poset and $T \subseteq S$ be a subset of $S$. An element $l \in S$ is a lower bound of $T$ if $l \sqsubseteq t$, for each $t \in T$. The lower bound $l$ of $T$ for which $l' \sqsubseteq l$ holds for every other lower bound $l'$ of $T$ is the infimum or the greatest lower bound of $T$, denoted by $\inf T$.*

When they exist, infima are unique by antisymmetry.

**Definition 8** (Semilattice). *A poset $(L, \sqsubseteq)$ is a* semilattice, *if for each $S \subseteq L$, $S$ has an infimum.*

**Definition 9** (Subsemilattice). *For a semilattice $(L, \sqsubseteq)$ and a set $S \subseteq L$, the poset $(S, \sqsubseteq)$ is a subsemilattice if $S$ is closed under infima, i.e., $\inf T \in S$ for each subset $T \subseteq S$.*

We next formalize Datalog inference using the immediate consequence operator. Here, we write $\theta$ for a substitution, i.e., a mapping from variables to constants in $\mathsf{U}$. The application $\alpha\theta$ of the substitution $\theta$ to the atom $\alpha$ replaces each variable $X$ in $\alpha$ with the constant $\theta(X)$.

**Definition 10.** *The immediate consequence operator $T_{\mathcal{P}} : 2^{\mathsf{B}} \to 2^{\mathsf{B}}$ adds the facts and all atoms that can be derived from an interpretation to that interpretation.*

$$T_{\mathcal{P}}(I) = I \cup \mathcal{F} \cup \{\mathsf{h}\theta \mid \mathsf{h} \leftarrow \mathsf{b_1}, \dots, \mathsf{b_n} \in \mathcal{R},$$
$$\theta \text{ is a substitution}, \mathsf{b_i}\theta \in I \text{ for all } i\}.$$

We slightly deviate from the standard definition by also including $I$ in $T_{\mathcal{P}}(I)$. But this will be useful later. Next, we define the fixed point operator, as we later use the fixed points of the immediate consequence operator.

**Definition 11** (Fixed Point Operator). *Given a set $S$, the fixed point operator $\mathsf{Fix} : (S \to S) \to 2^S$ maps a function $f : S \to S$ to to the set of fixed points of $f$.*

$$\mathsf{Fix}(f) = \{s \mid s \in S, f(s) = s\}$$

Besides regular Datalog models, we are interested in saturated models. Observe that the least model in Definition 2 is a saturated model.

**Definition 12** (Saturated Model). *For an equivalence relation $e_{\mathsf{U}} \in \mathcal{E}_{\mathsf{U}}$, a set of ground atoms $M \subseteq \mathsf{B}$ is a saturated model of $\mathcal{P}$ subject to $e_{\mathsf{U}}$ if the following two conditions hold:*

  *1. $M$ is a model, i.e., $M \models \mathcal{P}$.*

*2. M is a saturated set subject to $e_\mathsf{B}$.*

Finally, we also introduce Gram matrices, as they relate to our choice of factorization in Section 4.1.

**Definition 13** (Gram matrix). *Given $n$ vectors $\vec{v}_1, \vec{v}_2, \ldots, \vec{v}_n$, the Gram matrix $G$ is the matrix with as elements the inner products $G_{ij} = \langle \vec{v}_i, \vec{v}_j \rangle$.*

Every Gram matrix is symmetric and is positive semi-definite [Horn and Johnson, 2012, Theorem 7.2.10]. For real symmetric matrices, positive semi-definite means all eigenvalues are positive.

## A.2 Fixed point Semantics of (Saturated) Programs

We now use the above terminology to show that the models $\mathcal{M}(\mathcal{P})$ of a Datalog program $\mathcal{P}$ form a semilattice.

**Lemma 1.** *For any Datalog program $\mathcal{P}$, $(\mathcal{M}(\mathcal{P}), \subseteq)$ is a semilattice, where $\subseteq$ denotes set inclusion.*

*Proof.* According to Definition 8, $(\mathcal{M}(\mathcal{P}), \subseteq)$ is a semilattice if (1) it is a poset and (2) for each $S \subseteq \mathcal{M}(\mathcal{P})$, the poset $(S, \sqsubseteq)$ has an infimum. Regarding (1), $(\mathcal{M}(\mathcal{P}), \subseteq)$ is a poset since $\subseteq$ is a reflexive, antisymmetric, and transitive binary relation. To complete our proof, we need to show that (2) holds as well. The proof continues as follows. The intersection $M = \bigcap_i M_i$ of any set of models $\{M_i\} \subseteq \mathcal{M}(\mathcal{P})$ is also a model. This is because for any rule $\mathsf{h} \leftarrow \mathsf{b}_1, \ldots, \mathsf{b}_n \in \mathcal{R}$ and any mapping $\theta$, if all $\mathsf{b}_\mathsf{j}\theta \in M$, then all $\mathsf{b}_\mathsf{j}\theta \in M_i$ for all $i$. Since each $M_i$ is a model, $\mathsf{h}\theta \in M_i$ for all $i$, thus $\mathsf{h}\theta \in M$. Therefore, $M$ is a lower bound for $\{M_i\}$.

$M$ is the greatest lower bound, as for any lower bound $M'$ for $\{M_i\}$ it holds that $M' \subseteq M_i$ for each $i$. So $M' \subseteq \bigcap_i M_i = M$. This proves that every set of models has an infimum and hence $\mathcal{M}(\mathcal{P})$ is a semilattice. $\square$

As is typical in the Datalog literature [Ceri et al., 1989], we characterize the models of a Datalog program as the interpretations where nothing new is derived by the immediate consequence operator $T_\mathcal{P}$. The lemma below is a variant of the Knaster-Tarski theorem.

**Lemma 2.** *The set of fixed points of the immediate consequence operator $\mathsf{Fix}(T_\mathcal{P})$ equals the set of models $\mathcal{M}(\mathcal{P})$.*

$$\mathsf{Fix}(T_\mathcal{P}) = \mathcal{M}(\mathcal{P})$$

*Proof.* We prove the two directions $\mathsf{Fix}(T_\mathcal{P}) \subseteq \mathcal{M}(\mathcal{P})$ and $\mathsf{Fix}(T_\mathcal{P}) \supseteq \mathcal{M}(\mathcal{P})$ separately.

($\subseteq$) Let $I \in \mathsf{Fix}(T_\mathcal{P})$, so $T_\mathcal{P}(I) = I$. By Definition 10, $\mathcal{F} \subseteq T_\mathcal{P}(I) = I$. Moreover, for any rule $\mathsf{h} \leftarrow \mathsf{b}_1, \ldots, \mathsf{b}_n \in \mathcal{R}$ and grounding $\theta$, if $\mathsf{b}_\mathsf{i}\theta \in I$ for all $i$, we have $\mathsf{h}\theta \in T_\mathcal{P}(I) = I$. It follows that $I \in \mathcal{M}(\mathcal{P})$.

($\supseteq$) Let $M \in \mathcal{M}(\mathcal{P})$. Again by Definition 10, $T_\mathcal{P}(M) = M \cup \mathcal{F} \cup \{\mathsf{h}\theta \mid \ldots\}$. As $M$ is a model, $\mathcal{F} \subseteq M$. Moreover, for any rule $\mathsf{h} \leftarrow \mathsf{b}_1, \ldots, \mathsf{b}_n \in \mathcal{R}$ and grounding $\theta$, if $\mathsf{b}_\mathsf{i}\theta \in I$ for all $i$, we have $\mathsf{h}\theta \in M$ as $M$ is a model of $\mathcal{P}$. Thus we have $T_\mathcal{P}(M) = M$. $\square$

We denote the set of all saturated models subject to $e_\mathsf{U}$ as $\mathcal{M}_{e_\mathsf{U}}(\mathcal{P})$. By the above definition, every saturated model is also a model. Hence, it follows immediately that $\mathcal{M}_{e_\mathsf{U}}(\mathcal{P}) \subseteq \mathcal{M}(\mathcal{P})$.

Next, we clarify the relation between the regular models of a Datalog program and the saturated models of a program.

**Lemma 3.** *Let $\mathcal{P}$ be a Datalog program and $e_\mathsf{U} \in \mathcal{E}_\mathsf{U}$ be an equivalence relation. Then $(\mathcal{M}_{e_\mathsf{U}}(\mathcal{P}), \subseteq)$ is a subsemilattice of $(\mathcal{M}(\mathcal{P}), \subseteq)$.*

*Proof.* Let $\{M_i\} \subseteq \mathcal{M}_{e_\mathsf{U}}(\mathcal{P})$ be a subset. We prove that the intersection $M = \bigcap_i M_i$ is a saturated model and hence $\mathcal{M}_{e_\mathsf{U}}(\mathcal{P})$ is closed under infima. To do so, we verify the two conditions of Definition 12. 1) Due to Lemma 1, $M$ is a model of $\mathcal{P}$. 2) Consider a ground atom $\alpha \in M \cap \mathsf{B}$ and $e_\mathsf{B}(\alpha, \alpha')$ where $\alpha' \in \mathsf{B}$. Then $\alpha \in M_i$ for all $i$. Since each $M_i$ is saturated, $\alpha' \in M_i$ for all $i$. Therefore, $\alpha' \in M$. $\square$

As a corollary, the saturated program of Definition 2 always exists.

**Corollary 2.** *For each Datalog program $\mathcal{P}$ and each equivalence relation $e_\mathsf{U} \in \mathcal{E}_\mathsf{U}$, there exists a unique least saturated model:*

$$\inf \mathcal{M}_{e_\mathsf{U}}(\mathcal{P}) = \bigcap_{M \in \mathcal{M}_{e_\mathsf{U}}(\mathcal{P})} M \tag{11}$$

*Proof.* Since $\mathcal{M}_{e_\mathsf{U}}(\mathcal{P})$ is a subsemilattice (Lemma 3), the infimum of the set of all saturated models is also a saturated model. Uniqueness follows from antisymmetry of $\sqsubseteq$. $\qquad\square$

Finally, we show that the models of a saturated program correspond with the models of a program where the constants get instantiated.

**Lemma 4.** *For any equivalence relation $e_\mathsf{U} \in \mathcal{E}_\mathsf{U}$ and any program $\mathcal{P} = (\mathcal{F}, \mathcal{R})$, the semilattice $\mathcal{M}_{e_\mathsf{U}}(\mathcal{P})$ is isomorphic to the semilattice $\mathcal{M}((\mathsf{Inst}(\mathcal{F}, \mathbf{V}), \mathcal{R}))$.*

*Proof.* Let $\mathcal{P}' = (\mathsf{Inst}(\mathcal{F}, \mathbf{V}), \mathcal{R})$ be the instantiated program and $\mathsf{Inst} : \mathsf{U} \to \mathsf{U}/e_\mathsf{U}$ denote the map sending each constant to its equivalence class under $e_\mathsf{U}$, which we extend pointwise to atoms and sets of atoms:

$$\mathsf{Inst}(\mathsf{p}(\mathsf{t}_1, \ldots, \mathsf{t}_n), \mathbf{V}) = \mathsf{p}(\mathsf{Inst}(t_1, \mathbf{V}), \ldots, \mathsf{Inst}(t_n, \mathbf{V})), \quad \mathsf{Inst}(I, \mathbf{V}) = \{ \mathsf{Inst}(\alpha, \mathbf{V}) \mid \alpha \in I \}.$$

To prove the isomorphism, we first show that $\mathsf{Inst}$ is a bijection between $\mathcal{M}_{e_\mathsf{U}}(\mathcal{P})$ and $\mathcal{M}(\mathcal{P}')$ (by showing injectivity and surjectivity) and second show that infima are preserved by $\mathsf{Inst}$.

**Injection.** Let $M_1, M_2 \in \mathcal{M}_{e_\mathsf{U}}(\mathcal{P})$ such that $\mathsf{Inst}(M_1, \mathbf{V}) = \mathsf{Inst}(M_2, \mathbf{V})$. Take any atom $\alpha \in M_1$. By construction, $\mathsf{Inst}(\alpha, \mathbf{V}) \in \mathsf{Inst}(M_1, \mathbf{V}) = \mathsf{Inst}(M_2, \mathbf{V})$. This means that there is an atom $\alpha' \in M_2$ such that $\mathsf{Inst}(\alpha', \mathbf{V}) = \mathsf{Inst}(\alpha, \mathbf{V})$. By the definition of instantiation, this implies that $(\alpha, \alpha') \in e_\mathsf{B}$. But $M_2$ is saturated subject to $e_\mathsf{B}$, so it also holds that $\alpha \in M_2$. We can analogously prove that any atom $\alpha \in M_2$ is also an element of $M_1$, proving that $M_1 = M_2$.

**Surjection.** Take any instantiated model $M' \in \mathcal{M}(\mathcal{P}')$. Then define the model $M = \{\alpha \mid \mathsf{Inst}(\alpha, \mathbf{V}) \in M'\}$. Now it holds that $\mathsf{Inst}(M, \mathbf{V}) = \{\mathsf{Inst}(\alpha, \mathbf{V}) \mid \mathsf{Inst}(\alpha, \mathbf{V}) \in M'\} = M'$, meaning $\mathsf{Inst}$ is surjective.

**Preservation of infima.** Consider two saturated models $M_1, M_2 \in \mathcal{M}_{e_\mathsf{U}}(\mathcal{P})$. $\mathsf{Inst}$ commutes with intersections as

$$\begin{aligned} \mathsf{Inst}(M_1 \cap M_2, \mathbf{V}) &= \{\mathsf{Inst}(\alpha, \mathbf{V}) \mid \alpha \in M_1 \cap M_2\} \\ &= \{\mathsf{Inst}(\alpha, \mathbf{V}) \mid \alpha \in M_1\} \cap \{\mathsf{Inst}(\alpha, \mathbf{V}) \mid \alpha \in M_2\} \\ &= \mathsf{Inst}(M_1, \mathbf{V}) \cap \mathsf{Inst}(M_2, \mathbf{V}). \end{aligned}$$

Hence $\mathsf{Inst}$ is a bijection that preserves infima, so $(\mathcal{M}_{e_\mathsf{U}}(\mathcal{P}), \subseteq)$ and $(\mathcal{M}(\mathcal{P}'), \subseteq)$ are isomorphic semilattices. $\qquad\square$

### A.3 Main results

**Theorem 1.** *For any equivalence relation $e_\mathsf{U} \in \mathcal{E}_\mathsf{U}$, the program $(\mathcal{F} \cup \mathsf{F}(e_\mathsf{U}), \mathcal{R} \cup \mathsf{C}_{e_\mathsf{U}}(\mathcal{R}))$ is a saturation $\mathcal{P}_{|e_\mathsf{U}}$ of the program $\mathcal{P}$ subject to $e_\mathsf{U}$.*

*Proof.* Let $\mathcal{P}_{|e_\mathsf{U}} = (\mathcal{F} \cup \mathsf{F}(e_\mathsf{U}), \mathcal{R} \cup \mathsf{C}_{e_\mathsf{U}}(\mathcal{R}))$. According to Definition 2, to prove that $\mathcal{P}_{|e_\mathsf{U}}$ is a saturation of $\mathcal{P}$ we need to show that $\mathsf{M}(\mathcal{P}_{|e_\mathsf{U}}) \cap \mathsf{B}$ is the smallest model of $\mathcal{P}$ that is saturated subject to $e_\mathsf{B}$. To prove this, we first show that $\{M \cap \mathsf{B} \mid M \in \mathcal{M}(\mathcal{P}_{|e_\mathsf{U}})\} = \mathcal{M}_{e_\mathsf{U}}(\mathcal{P})$, by checking the two directions ($\subseteq$ and $\supseteq$).

($\subseteq$) Let $M \in \mathcal{M}(\mathcal{P}_{|e_\mathsf{U}})$. We show that $M \cap \mathsf{B} \in \mathcal{M}_{e_\mathsf{U}}(\mathcal{P})$ by verification of the two conditions of Definition 12.

1. Given that $\mathcal{F} \subseteq \mathcal{F} \cup \mathsf{F}(e_\mathsf{U})$ and $\mathcal{R} \subseteq \mathcal{R} \cup \mathsf{C}_{e_\mathsf{U}}(\mathcal{R})$, every atom entailed in $\mathcal{P}$ is also entailed in $\mathcal{P}_{|e_\mathsf{U}}$ due to monotonicity. Therefore, $M \cap \mathsf{B}$ is also a model of $\mathcal{P}$.

2. Let $\alpha = \mathsf{p}(\mathsf{t}_1, \ldots, \mathsf{t}_\mathsf{n}) \in M \cap \mathsf{B}$ and suppose $e_\mathsf{B}(\alpha, \alpha')$ where $\alpha' = \mathsf{p}(\mathsf{t}'_1, \ldots, \mathsf{t}'_\mathsf{n}) \in \mathsf{B}$. The congruence rule (4) for predicate $\mathsf{p}$ grounded by $\mathsf{X}_i \mapsto \mathsf{t}'_i$ and $\mathsf{X}_i \mapsto \mathsf{t}'_i$ is

$$\mathsf{p}(\mathsf{t}'_1, \ldots, \mathsf{t}'_\mathsf{n}) \leftarrow \mathsf{p}(\mathsf{t}_1, \ldots, \mathsf{t}_\mathsf{n}) \wedge \bigwedge_{i=1}^n \mathsf{t}'_i \overset{e_\mathsf{U}}{\approx} \mathsf{t}_i.$$

By Equation 2, $e_\mathsf{B}(\alpha, \alpha')$ implies $e_\mathsf{U}(\mathsf{t}_i, \mathsf{t}'_i)$ for all $i \in \{1, \ldots, n\}$. Since $e_\mathsf{U}(\mathsf{t}_i, \mathsf{t}'_i)$, we have $\mathsf{t}_i \overset{e_\mathsf{U}}{\approx} \mathsf{t}'_i \in \mathsf{F}(e_\mathsf{U}) \subseteq M$. So as $M$ is a model, $\mathsf{p}(\mathsf{t}'_1, \ldots \mathsf{t}'_\mathsf{n}) \in M$. Therefore, $M$ is saturated.

$(\supseteq)$ Let $M \in \mathcal{M}_{e_\mathsf{U}}(\mathcal{P})$. We show $M \cup \mathsf{F}(e_\mathsf{U}) \in \mathcal{M}(\mathcal{P}_{|e_\mathsf{U}})$, by verifying that $M$ is satisfies all rules in $\mathcal{R} \cup \mathsf{C}_{e_\mathsf{U}}(\mathcal{R})$ and contains all facts in $\mathcal{F} \cup \mathsf{F}(e_\mathsf{U})$. First, since $M$ is a model of $\mathcal{P}$, we have $M$ contains all facts $\mathcal{F}$ and satisfies all rules $\mathcal{R}$. Lastly, consider a congruence rule in $\mathsf{C}_{e_\mathsf{U}}(\mathcal{R})$

$$\mathsf{p}(\mathsf{X}_1, \ldots, \mathsf{X}_\mathsf{n}) \leftarrow \mathsf{p}(\mathsf{X}'_1, \ldots, \mathsf{X}'_\mathsf{n}) \wedge \bigwedge_{i=1}^n \mathsf{X}_i \overset{e_\mathsf{U}}{\approx} \mathsf{X}'_i$$

and a grounding $\theta$ with $\mathsf{X}_i \mapsto \mathsf{t}_i$ and $\mathsf{X}'_i \mapsto \mathsf{t}'_i$. Suppose all body atoms are in $M$: $\mathsf{p}(\mathsf{t}'_1, \ldots, \mathsf{t}'_\mathsf{n}) \in M \cap \mathsf{B}$ and $\mathsf{t}_i \overset{e_\mathsf{U}}{\approx} \mathsf{t}'_i \in M$ for all $i$. Since $\mathsf{t}_i \overset{e_\mathsf{U}}{\approx} \mathsf{t}'_i \in \mathsf{F}(e_\mathsf{U})$, we have $e_\mathsf{U}(\mathsf{t}_i, \mathsf{t}'_i)$ for all $i$. By Equation 2, this means $e_\mathsf{B}(\mathsf{p}(\mathsf{t}_1, \ldots, \mathsf{t}_\mathsf{n}), \mathsf{p}(\mathsf{t}'_1, \ldots, \mathsf{t}'_\mathsf{n}))$. Since $\mathsf{p}(\mathsf{t}'_1, \ldots, \mathsf{t}'_\mathsf{n}) \in M \cap \mathsf{B}$ and $M$ is saturated, we have $\mathsf{p}(\mathsf{t}_1, \ldots, \mathsf{t}_\mathsf{n}) \in M$. Therefore, $M$ is closed under all congruence rules.

We have shown $\{M \cap \mathsf{B} \mid M \in \mathcal{M}(\mathcal{P}_{|e_\mathsf{U}})\} = \mathcal{M}_{e_\mathsf{U}}(\mathcal{P})$. In other words, the models of $\mathcal{P}_{|e_\mathsf{U}}$ without the equality facts are precisely the saturated models of $\mathcal{P}$. The theorem by Corollary 2, as the least saturated model $\inf \mathcal{M}_{e_\mathsf{U}}(\mathcal{P})$ exists and equals the least model of $\mathcal{P}'$. $\square$

**Theorem 2.** *For each possible world $w \subseteq \mathsf{F}(\rho)$ of a soft unification program $\mathcal{P}_\mathbf{s}$ and each similarity function $d$ satisfying*

$$d(\vec{v}_1, \vec{v}_2) = 1 \text{ if and only if } \vec{v}_1 = \vec{v}_2, \forall \vec{v}_1, \vec{v}_2 \in \mathbb{R}^k, \tag{6}$$

*$P(w) = 1$ implies that the $\approx$-facts in $w$ satisfy the semantics of transitivity.*

*Proof.* Consider three constants $\mathsf{a}, \mathsf{b}, \mathsf{c} \in \mathsf{U}$, such that $(\mathsf{a} \approx \mathsf{b}) \in w$ and $(\mathsf{b} \approx \mathsf{c}) \in w$. Because $P(w) = 1$, it follows that $P(\mathsf{a} \approx \mathsf{b}) = 1$ and $P(\mathsf{b} \approx \mathsf{c}) = 1$, meaning that $\vec{v}_\mathsf{a} = \vec{v}_\mathsf{b}$ and $\vec{v}_\mathsf{b} = \vec{v}_\mathsf{c}$. Since $P(w) = 1$, it follows that $P(\mathsf{a} \approx \mathsf{c}) = 1$ and hence, $(\mathsf{a} \approx \mathsf{c}) \in w$. $\square$

**Theorem 3.** *The probability $P(\alpha)$ a ground atom $\alpha \notin \mathcal{F}$ is true in a soft unification program $\mathcal{P}_\mathbf{s}$ is not a multilinear polynomial function in the embeddings.*

*Proof.* As defined in (1), the probability an atom in a probabilistic logic program $\mathcal{P}_\mathbf{p}$ is

$$P(\alpha) = \sum_{\substack{w \in 2^\mathcal{F} \\ (w, \mathcal{R}) \models \alpha}} \prod_{\mathsf{f} \in w} P_f(\mathsf{f}) \prod_{\mathsf{f} \in \mathcal{F} \setminus w} (1 - P_f(\mathsf{f})).$$

In the case of soft unification program $\mathcal{P}_\mathbf{s}$, this becomes

$$P(\alpha) = \sum_{\substack{w \in 2^{\mathsf{F}(\rho)} \\ (\mathcal{F} \cup w, \mathcal{R} \cup \mathsf{C}_\rho(\mathcal{R})) \models \alpha}} \prod_{\mathsf{a} \approx \mathsf{b} \in w} d(\vec{v}_\mathsf{a}, \vec{v}_\mathsf{b}) \prod_{\mathsf{a} \approx \mathsf{b} \in \mathsf{F}(\rho) \setminus w} (1 - d(\vec{v}_\mathsf{a}, \vec{v}_\mathsf{b})).$$

As soon as there are more than two constants in the Herbrand universe, the vector $\vec{v}_\mathsf{c}$ for each constant $\mathsf{c}$ occurs multiple times in the above product. This implies $P(\alpha)$ is not multilinear.

As an example, consider a program with facts $\{\mathsf{p}(\mathsf{a}), \mathsf{q}(\mathsf{b})\}$ and the rules $\{r(X) :- p(X), q(X)\}$. If we query $\mathsf{r}(\mathsf{c})$ in this program we get $P(\mathsf{r}(\mathsf{c})) = d(\vec{v}_\mathsf{a}, \vec{v}_\mathsf{c}) \cdot d(\vec{v}_\mathsf{b}, \vec{v}_\mathsf{c})$, because $\mathsf{a} \approx \mathsf{b}$ and $\mathsf{b} \approx \mathsf{c}$ allow us to conclude $\mathsf{p}(\mathsf{c})$ and $\mathsf{q}(\mathsf{c})$, respectively. If the function $d$ is for instance the inner product, then it follows that $P(\mathsf{r}(\mathsf{c})) = \langle \vec{v}_\mathsf{a}, \vec{v}_\mathsf{c} \rangle \cdot \langle \vec{v}_\mathsf{b}, \vec{v}_\mathsf{c} \rangle$, which is not linear in $\vec{v}_\mathsf{c}$. $\square$

**Theorem 4.** *Consider a matrix $M$ that contains all the marginal probabilities of constants being equivalent under the probabilistic equivalence semantics, i.e., $M_{i,j} = P(\mathsf{c}_i \approx \mathsf{c}_j)$ with $\mathsf{c}_i, \mathsf{c}_j \in \mathsf{U}$. If $P_\mathcal{E}$ is factorized as in (9), $M$ is positive semi-definite.*

*Proof.* As shown in Section 4.1, under the factorization of Equation 9, every element $M_{i,j}$ is an inner product between the probability vectors of $\mathsf{c}_i$ and $\mathsf{c}_j$. In other words, $M$ is a Gram matrix, which is positive semi-definite [Horn and Johnson, 2012, Theorem 7.2.10]. $\square$

**Theorem 5.** *For each distribution $P_{\mathcal{E}} : \mathcal{E}_{\mathsf{U}} \to [0,1]$ and each target ground p-atom $\alpha$, we have*

$$\mathbb{E}_{e_{\mathsf{U}} \sim P_{\mathcal{E}}} \left[ \mathcal{P}_{|e_{\mathsf{U}}} \models \alpha \right] = \mathbb{E}_{e_{\mathsf{U}} \sim P_{\mathcal{E}}} [(\mathcal{F} \cup \mathsf{F}(e_{\mathsf{U}}), \mathsf{Mag}(\mathsf{Sg}(\mathcal{R}, \mathsf{p}), \alpha)) \models \alpha].$$

*Proof.* To prove this theorem, it suffices to show that for each $e_{\mathsf{U}} \in P_{\mathcal{E}}$, $(\mathcal{F} \cup \mathsf{F}(e_{\mathsf{U}}), \mathsf{Mag}(\mathsf{Sg}(\mathcal{R}, \mathsf{p}), \alpha))$ entails $\alpha$ if and only if $\mathcal{P}_{|e_{\mathsf{U}}}$ entails $\alpha$.

We start be recalling Theorem 1, which says that

$$\mathcal{P}_{|e_{\mathsf{U}}} \models \alpha \Leftrightarrow (\mathcal{F} \cup \mathsf{F}(e_{\mathsf{U}}), \mathcal{R} \cup \mathsf{C}(\mathcal{R})) \models \alpha.$$

On the right side, we have a regular Datalog program. So as singularization preserves entailment [Marnette, 2009], it holds that

$$\mathcal{P}_{|e_{\mathsf{U}}} \models \alpha \Leftrightarrow (\mathcal{F} \cup \mathsf{F}(e_{\mathsf{U}}), \mathsf{Sg}(\mathcal{R}, \mathsf{p})) \models \alpha.$$

Furthermore, due to the soundness of the magic sets transformation [Beeri and Ramakrishnan, 1987, Theorem 4.1], it also holds that

$$\mathcal{P}_{|e_{\mathsf{U}}} \models \alpha \Leftrightarrow (\mathcal{F} \cup \mathsf{F}(e_{\mathsf{U}}), \mathsf{Mag}(\mathsf{Sg}(\mathcal{R}, \mathsf{p}), \alpha))$$

As the above equivalence holds for any equivalence relation $e_{\mathsf{U}}$, it also holds in expectation, and the theorem follows. $\square$

**Theorem 6.** *For each distribution $P_{\mathcal{E}} : \mathcal{E}_{\mathsf{U}} \to [0,1]$ and each target ground p-atom $\alpha$, we have*

$$\mathbb{E}_{e_{\mathsf{U}} \sim P_{\mathcal{E}}} \left[ \mathcal{P}_{|e_{\mathsf{U}}} \models \alpha \right] = \lim_{k \to \infty} \frac{1}{k} \sum_{i=1}^{k} \mathbb{1}[(\mathsf{Inst}(\mathcal{F}, \mathbf{V}_i), \mathsf{Mag}(\mathcal{R}, \alpha)) \models \mathsf{Inst}(\alpha, \mathbf{V}_i)] , \text{ where } \mathbf{V}_i \sim P_{\mathcal{E}}.$$

$$(10)$$

*Proof.* From the isomorphism of Lemma 4, it follows that

$$\mathcal{P}_{|e_{\mathsf{U}}} \models \alpha \quad \Leftrightarrow \quad (\mathsf{Inst}(\mathcal{F}, \mathbf{V}), \mathcal{R}) \models \mathsf{Inst}(\alpha, \mathbf{V}).$$

Furthermore, due to the correctness of the magic sets transformation [Beeri and Ramakrishnan, 1987, Theorem 4.1], it also holds that

$$\mathcal{P}_{|e_{\mathsf{U}}} \models \alpha \quad \Leftrightarrow \quad (\mathsf{Inst}(\mathcal{F}, \mathbf{V}), \mathsf{Mag}(\mathcal{R}, \alpha)) \models \mathsf{Inst}(\alpha, \mathbf{V}).$$

Hence, in expectation, we also get

$$\mathbb{E}_{e_{\mathsf{U}}} \left[ \mathcal{P}_{|e_{\mathsf{U}}} \models \alpha \right] = \mathbb{E}_{e_{\mathsf{U}}} \left[ (\mathsf{Inst}(\mathcal{F}, \mathbf{V}), \mathsf{Mag}(\mathcal{R}, \alpha)) \models \mathsf{Inst}(\alpha, \mathbf{V}) \right].$$

Finally, we can apply the law of large numbers to arrive at the theorem.

$$\mathbb{E}_{e_{\mathsf{U}}} \left[ \mathcal{P}_{|e_{\mathsf{U}}} \models \alpha \right] = \lim_{k \to +\infty} \frac{1}{k} \sum_{i=1}^{k} \mathbb{1}[(\mathsf{Inst}(\mathcal{F}, \mathbf{V}_i), \mathsf{Mag}(\mathcal{R}, \alpha)) \models \mathsf{Inst}(\alpha, \mathbf{V}_i)].$$

$$\square$$

# B Experimental Details & Hyperparameters

**Hyperparameters.** For the countries knowledge graph and the differentiable finite state machines, we use similar hyperparameters as DeepSoftLog, see Table 5 and Table 7. In all experiments, we used 100 samples for the gradient estimation and batch size 1 as this provided a sufficient training signal. For the nations knowledge graph, we performed Bayesian tuning on the validation set, optimizing for the MRR (see Table 6).

**Program Templates** The same program templates as the NTP and DeepSoftLog are used. For Countries S1, we have

$r_1(X, Y) \leftarrow r_2(Y, X)$
$r_1(X, Y) \leftarrow r_2(X, Z) \wedge r_2(Z, Y)$

For Countries S2, we have

$r_1(X, Y) \leftarrow r_2(Y, X)$
$r_1(X, Y) \leftarrow r_2(X, Z) \wedge r_2(Z, Y)$
$r_1(X, Y) \leftarrow r_2(X, Z) \wedge r_3(Z, Y)$

For Countries S3, we have

$r_1(X, Y) \leftarrow r_2(Y, X)$
$r_1(X, Y) \leftarrow r_2(X, Z) \wedge r_2(Z, Y)$
$r_1(X, Y) \leftarrow r_2(X, Z) \wedge r_3(Z, Y)$
$r_1(X, Y) \leftarrow r_2(X, A) \wedge r_3(B, Y) \wedge r_4(A, B)$

For nations, the following templates are repeated 20 times each.

$r_1(X, Y) \leftarrow r_2(X, Y)$
$r_1(X, Y) \leftarrow r_1(X, Y)$
$r_1(X, Y) \leftarrow r_2(X, Z) \wedge r_3(Z, Y).$

For the differentiable finite state machines, we simply implement a finite state in datalog. Datalog does not support functors, meaning that the finite state implementation has a fixed max length on the inputs it accepts. As we only test on inputs of max length 8 this is not a problem, however.

$\mathsf{accepts}(S1, S2, S3, S4, S5, S6, S7, S8, S9) \leftarrow \mathsf{run}(\mathsf{end\_state}, S1, S2, S3, S4, S5, S6, S7, S8, S9)$

$\mathsf{run}(\mathsf{STATE}, \mathsf{SYMBOL}, S1, S2, S3, S4, S5, S6, S7, S8) \leftarrow$
$\qquad\qquad\qquad \mathsf{run}(\mathsf{OLDSTATE}, S1, S2, S3, S4, S5, S6, S7, S8, -1)$
$\qquad\qquad\qquad \mathsf{transition}(\mathsf{OLDSTATE}, \mathsf{STATE}, \mathsf{SYMBOL})$

**Regularization.** The nations knowledge graph quickly exhibited overfitting. Proper regularization is often crucial to obtain competitive results on link prediction [Lacroix et al., 2018]. As we have a probabilistic model, we can simply regularize the entropy, meaning we minimize the entropy of the latent random variables. In the differentiable finite state machines we regularize the neural network using weight decay.

**Data License.** The countries (ODbL licence) and nations (CC0 license) knowledge graphs, grammars (CC0 license), and MNIST dataset (MIT license) are all publicly available.

**Compute.** We performed the experiments on servers with an Intel i7-12700 CPU and 64GB RAM, although lower resources may suffice. No GPU or TPU compute was used. Roughly speaking, one differentiable finite state machine experiment takes 3 minutes, one countries experiment takes 3 to 5 hours (depending on S1/S2/S3), and one nations experiment takes about 1 hour.

**Learned Rules** Some example of learned rules, for countries (S2):

$\mathsf{neighborOf}(X, Y) \leftarrow \mathsf{neighborOf}(Y, X)$
$\mathsf{locatedIn}(X, Y) \leftarrow \mathsf{locatedIn}(X, Z), \mathsf{locatedIn}(Z, Y)$

And for the nations knowledge graph:

$\mathsf{relexports}(X, Y) \leftarrow \mathsf{booktranslations}(X, Y)$
$\mathsf{conferences}(X, Y) \leftarrow \mathsf{conferences}(Y, X)$
$\mathsf{embassy}(X, Y) \leftarrow \mathsf{warning}(X, Z), \mathsf{commonbloc2}(Z, Y)$

**Additional Experiments.** In Table 8, we repeat the knowledge graph experiments on the Kinship and UMLS datasets. We apply the same evaluation protocol and use the dataset splits from Qu et al. [2021]. Hyperparameters are again selected by tuning on the validation MRR for the UMLS dataset. For the Kinship dataset, we simply reuse these hyperparameters.

Table 5: Hyperparameters used in the countries experiment (c.f. Table 1). Identical hyperparameters are used for the three different variants (S1, S2, and S3).

| Hyperparameter name | Value |
| --- | --- |
| Optimizer | AdamW |
| Learning rate | 0.1 |
| Number of samples | 100 |
| Number of epochs | 1 |
| Batch size | 1 |

Table 6: Hyperparameters used in the nations experiment (c.f. Table 1).

| Hyperparameter name | Value |
| --- | --- |
| Optimizer | AdamW |
| Learning rate | 0.0013 |
| Number of samples | 100 |
| Number of epochs | 2 |
| Batch size | 1 |
| Gradient Clipping | 0.000093 |
| Entropy regularization | 0.0000025 |
| Max fixed-point iterations | 5 |

## C Linking Equivalence and Possible World Semantics

The probabilistic equality and possible world semantics that have been described in the main paper are orthogonal to each other and can be combined by having a distribution both over the facts and over equivalence relations.

$$P(\alpha) = \mathbb{E}_{e_U, w}[(w, \mathcal{R})_{|e_U} \models \alpha]$$

For example, consider a program with the facts $\{r(a), r(b)\}$ and with no rules. Now, we both impose a probability distribution on the facts $\{r(a) \mapsto 0.2, r(b) \mapsto 0.8\}$ and the equivalence $P(a \approx b) = 0.5$. Then, if we query $r(a)$, we get

$$P(r(a)) = P(a \approx b) \cdot (P(r(a)) + (1 - P(r(a))) \cdot P(r(b))) + (1 - P(a \approx b)) \cdot P(r(a))$$
$$= 0.5 \cdot (0.2 + 0.8 \cdot 0.8) + 0.5 \cdot 0.2$$
$$= 0.52$$

On the other hand, we can also interpret the equivalence as a specific low-rank parameterization of the possible world semantics.

Table 7: Hyperparameters used in the differentiable finite state machines experiment (c.f. Table 2). Identical hyperparameters are used for the three different languages.

| Hyperparameter name | Value |
| --- | --- |
| Optimizer | AdamW |
| Learning rate (embeddings) | 0.1 |
| Learning rate (neural network) | 0.0001 |
| Number of samples | 100 |
| Number of epochs | 25 |
| Batch size | 1 |
| Weight Decay | 0.01 |

Table 8: Link prediction results on the UMLS and Kinship knowledge graphs. Hyperparameters are selected by tuning on the validation MRR, and we report the mean and standard deviation over 10 random seeds. Splits and CTP baseline results are taken from Qu et al. [2021].

| Datasets | Metrics | CTP | Ours |
|---|---|---|---|
| **Kinship** | MRR | 0.335 | **0.408** |
| | Hits@1 | 0.177 | **0.234** |
| | Hits@3 | 0.376 | **0.483** |
| | Hits@10 | 0.703 | **0.795** |
| **UMLS** | MRR | 0.404 | **0.557** |
| | Hits@1 | 0.288 | **0.392** |
| | Hits@3 | 0.430 | **0.666** |
| | Hits@10 | 0.674 | **0.880** |

$$
\begin{aligned}
P(\alpha) &= \mathbb{E}_{e_{\mathsf{U}} \sim P_{\mathcal{E}}} \left[ \mathcal{P}_{|e_{\mathsf{U}}} \models \alpha \right] && \text{By Definition 4.} \\
&= \mathbb{E}_{e_{\mathsf{U}} \sim P_{\mathcal{E}}} \left[ (\mathcal{F} \cup \mathsf{F}(e_{\mathsf{U}}), \mathcal{R} \cup \mathsf{C}_{e_{\mathsf{U}}}(\mathcal{R})) \models \alpha \right] && \text{By Theorem 1.} \\
&= \mathbb{E}_w \left[ (\mathcal{F} \cup w, \mathcal{R} \cup \mathsf{C}(\mathcal{R})) \models \alpha \right]
\end{aligned}
$$

In the last line, the possible world distribution is defined such that $P(w) = P(e_{\mathsf{U}})$ when there exists $e_{\mathsf{U}}$ with $w = \mathsf{F}(e_{\mathsf{U}})$ and 0 otherwise. Observe that this distribution is not factorized into independent facts, as is usual in probabilistic logic programming.

## D  Magic Sets Transformation

As the equality atoms do not appear in any rule heads, there is no need to specialize the magic towards equality as was done by [Benedikt et al., 2018]. Instead, we perform a standard magic sets transformation We demonstrate this on the running example below, without differentiating between the free and bound adornments.

**Example 7.** *We present the program transformation* $\mathsf{Mag}(\mathsf{Sg}(\mathcal{R}_{ex}, \mathsf{q}), \mathsf{q}(\mathsf{a}, \mathsf{c}))$ *of the rules in the example program* $\mathcal{P}_{ex}$ *of Example 2. We introduce the special relation* magic *to guard the execution of* r, *and* q *as the target relation. To query an atom of* r, *for instance* r(a, c), *we simply add the fact* magic(a, c) *to the program and query* q(a, c). *Note that the magic and singularization transforms are not unique, due to the different possible orderings of atoms.*

$\mathsf{r}(\mathsf{X}, \mathsf{Y}) \leftarrow \mathsf{magic}(\mathsf{X}, \mathsf{Y}) \wedge \mathsf{r}(\mathsf{X}, \mathsf{Z}_1) \wedge \mathsf{Z}_1 \approx \mathsf{Z}_2 \wedge \mathsf{r}(\mathsf{Z}_2, \mathsf{Y})$
$\mathsf{magic}(\mathsf{X}, \mathsf{Z}_1) \leftarrow \mathsf{r}(\mathsf{X}, \mathsf{Y}).$
$\mathsf{magic}(\mathsf{Z}_2, \mathsf{Y}) \leftarrow \mathsf{r}(\mathsf{X}, \mathsf{Y}) \wedge \mathsf{r}(\mathsf{X}, \mathsf{Z}_1) \wedge \mathsf{Z}_1 \approx \mathsf{Z}_2$
$\mathsf{q}(\mathsf{X}, \mathsf{Y}) \leftarrow \mathsf{magic}(\mathsf{X}, \mathsf{Y}) \wedge \mathsf{X} \approx \mathsf{X}_1 \wedge \mathsf{Y} \approx \mathsf{Y}_1 \wedge \mathsf{r}(\mathsf{X}_1, \mathsf{Y}_1)$

