# OpenReview forum: "Embeddings as Probabilistic Equivalence in Logic Programs"
_NeurIPS.cc/2025/Conference — NeurIPS 2025 poster_

### Official Review · Reviewer_uR2S · 2025-06-21

**Clarity:** 3
**Significance:** 2
**Originality:** 2
**Rating:** 4
**Confidence:** 4

**Summary:**

Soft unification has been used in neursoymbolic learning as a way to translate the symbolic comparison between symbols into an embedding space comparison. However, soft unification exploits symmetry, rather than equivalence, which can lead to facts being derived that cannot be true under equivalence (violating transitivity). This paper proposes a probabilistic semantics of equivalence for logic programming. In this new semantics, symbol partitions (groupings) can be learned, and transitivity allows new facts over the grouped symbols to be derived. The evaluation demonstrates that this approach generally leads to higher AUPRC, MRR, and HITS@m compared to the baselines across several benchmarks.

**Questions:**

- How does this new learning algorithm compare to DeepSoftLog on some of its scalability benchmarks, especially since this work is directly building on it? I would have liked to see a comparison for MNIST-Addition and Visual Sudoku. Even if the new notion of equivalence doesn’t improve results, it would be helpful to show whether there is any efficiency improvement or penalty, or any change in accuracy.
- Other than isIn/locatedIn, what are other examples of equivalences that were exploited in the evaluated benchmarks, resulting in performance improvements?
- What do the authors see as the  real-world application of this work? How does this work fit in with foundation models, which could likely eliminate the need for training the neural network?

**Ethical Concerns:**

["NO or VERY MINOR ethics concerns only"]

**Final Justification:**

After the rebuttal, the authors provided some insight into the runtime improvements of the proposed technique over the baselines. Additional information was provided about why equivalence is useful in some of the benchmarks. I can see the significance of this work in the field of neurosymbolic logic programming, but I am still at the borderline in terms of its general applicability.

**Limitations:**

yes

**Quality:**

3

**Strengths And Weaknesses:**

**Strengths:**

- This paper was well-written and easy to follow. The running examples were helpful in understanding the problem and proposed solution.
- The problem is well-motivated. This paper clearly outlines the weaknesses of prior work that reasons on embeddings in logic programs.
- The experiments demonstrate that exploiting symmetries between equivalent symbols substantially improves performance, especially for the differentiable automata benchmark.

**Weaknesses:**
- There are no experiments to compare the training efficiency of the different approaches. Some analysis (such as Figure 2 in [1]) would have been useful.
- This paper seems to be a direct improvement on DeepSoftLog, but several benchmarks from the DeepSoftLog paper were omitted, including MNIST-addition and visual sudoku classification.
- Minor comments:
    - Define “symbol” upfront. I have seen some neurosymbolic literature that uses “symbol” to mean the likely discrete neural network prediction. But this paper is using it to mean the name of a rule.
    - Line 59: “First, formalize” -> “First, we formalize”
    - Line 127: $P_ex$ -> $P_{ex}$

[1] Differentiable Reasoning on Large Knowledge Bases and Natural Language, AAAI 2020

---

> ### Author Rebuttal · Authors · 2025-07-30
>
> We sincerely thank the reviewer for taking the time to review our paper and providing useful feedback.
>
> > There are no experiments to compare the training efficiency of the different approaches.
>
> Our experiments indeed focus on accuracy improvements, rather than runtime performance. However, the runtimes of our experiments are generally faster than those of baselines, such as DeepSoftLog. For example, for the Countries S3 runs, the runtime was on average two hours per experiment, while for DeepSoftLog, this was five hours. Moreover, DeepSoftLog cannot handle e.g. the Nations experiment, due to scalability issues.
>
> > How does this new learning algorithm compare to DeepSoftLog on some of its scalability benchmarks, especially since this work is directly building on it? I would have liked to see a comparison for MNIST-Addition and Visual Sudoku.
>
> The MNIST-addition and Visual Sudoku experiments are less interesting in our setting as they are ground problems over linear arithmetic. Furthermore, the MNIST-addition encoding of DeepSoftLog does not fully use soft unification. So performing this experiment using our approach would give functionally identical results to DeepSoftLog.
>
> We have included experiments beyond the scalability limits of DeepSoftLog, namely on the Nations knowledge graph. Below, we furthermore repeated this last experiment on two other knowledge graphs (UMLS and Kinship), which are further out of reach for DeepSoftLog.
>
> | Datasets   | Metrics  | CTP   | Ours   |
> |------------|----------|--------|--------|
> | **Kinship**| MRR      | 0.335  | **0.408** |
> |            | Hits@1   | 0.177  | **0.234** |
> |            | Hits@3   | 0.376  | **0.483** |
> |            | Hits@10  | 0.703  | **0.795** |
> | **UMLS**   | MRR      | 0.404  | **0.557** |
> |            | Hits@1   | 0.288  | **0.392** |
> |            | Hits@3   | 0.430  | **0.666** |
> |            | Hits@10  | 0.674  | **0.880** |
>
> > Define “symbol” upfront. I have seen some neurosymbolic literature that uses “symbol” to mean the likely discrete neural network prediction. But this paper is using it to mean the name of a rule.
>
> In our context, a symbol is a constant in a Datalog program (i.e. an element of the Herbrand universe, see the second paragraph of Section 2). The outputs of a neural network can indeed be used as a distribution over symbols for a neurosymbolic model . In our work, however, symbols are embedded into a vector space.
>
> > Other than isIn/locatedIn, what are other examples of equivalences that were exploited in the evaluated benchmarks, resulting in performance improvements?
>
> Another example is in the differentiable finite state machine experiment, where symbols represent states in an automaton. Now, consider the following transitions between states: $a\rightarrow b$, $c\rightarrow d$, and $e\rightarrow f$. Assuming we have that $b=c$ and $c=e$, it should be possible to transition $a\rightarrow b (=c) \rightarrow d$ and $a\rightarrow b (=e) \rightarrow f$. However, the latter derivation cannot be found without transitivity. Table 3 demonstrates that the more faithful modelling of automata behaviour has major performance implications. More generally, this problem is encountered in any task where true equivalence is present (in this example, automata states).
>
> > What do the authors see as the real-world application of this work? How does this work fit in with foundation models, which could likely eliminate the need for training the neural network?
>
> Although our contribution is on the more fundamental side, we hope that it will enable future applications where symbolic and neural structures are jointly optimized, leading to models with interpretable reasoning traces that are critical in highly regulated domains (e.g., finance). Moreover, neurosymbolic models can provide formal guarantees about their behaviour, as previously demonstrated on applications with safety requirements (e.g., self-driving [1] or language model toxicity [2]).
>
> Foundation models have indeed altered the conventional machine learning paradigm, but still face limitations in these aspects. Furthermore, there are still questions about how far out-of-distribution foundation models can perform compositional reasoning (as a symbolic model can). For example, Yue et al. [3] argue that the currently popular RL-trained reasoning models do not fundamentally expand their capabilities beyond the pre-training distribution, and Hazra et al. [4] argue that foundation models still struggle with computationally hard inferences.
>
> [1] Giunchiglia, Eleonora, et al. "ROAD-R: the autonomous driving dataset with logical requirements." _Machine Learning_ (2023).
> [2] Ahmed, Kareem, Kai-Wei Chang, and Guy Van den Broeck. "A pseudo-semantic loss for autoregressive models with logical constraints." _NeurIPS_ (2023).
> [3] Yue, Yang, et al. "Does reinforcement learning really incentivize reasoning capacity in LLMs beyond the base model?." _ICML AI4Math workshop_ (2025).
> [4] Hazra, Rishi, et al. "Have Large Language Models Learned to Reason? A Characterization via 3-SAT Phase Transition. _COLM_ (2025).

---

> > ### Comment · Reviewer_uR2S · 2025-08-04
> >
> > Thank you for the detailed response and including experiments on additional knowledge graphs. I'm pleased to see that there are runtime improvements over DeepSoftLog. Also, the automata example is very helpful in understanding why equivalence can be useful. I suggest that the authors provide a similar explanation for each benchmark in the revised version.
> >
> > I'm still struggling to see the broader significance of this work. My comment about foundation models was not meant to imply that they can perform compositional reasoning well. What I meant is that there is no point in training the neural network from scratch when foundation models can easily classify MNIST digits (as in the automata benchmark), and these predictions can be run through the symbolic program. However, I can still see the significance of this work in the field of neurosymbolic logic programming, so I'll raise my score accordingly.

---

> > > ### Author Response · Authors · 2025-08-06
> > >
> > > >  I'm still struggling to see the broader significance of this work. My comment about foundation models was not meant to imply that they can perform compositional reasoning well. What I meant is that there is no point in training the neural network from scratch when foundation models can easily classify MNIST digits (as in the automata benchmark), and these predictions can be run through the symbolic program.
> > >
> > > In some neurosymbolic benchmarks, the neural component is kept very simple. So a foundation model could indeed trivially classify e.g. the MNIST digits in the automata benchmark. However, more generally, it is not the case that all classification tasks can be perfectly solved by a pretrained foundation model. It can hence be required to finetune a decent pretrained model to align its learned representations with the differentiable reasoning objective. See [1] for an example of this. Finally, we also note that probabilistic inference is also useful at inference time as simply running the symbolic program on the thresholded neural network predictions does not give the MAP solution and provides no uncertainty estimation.
> > >
> > > [1]: Huang, Jiani, et al. _"LASER: A Neuro-Symbolic Framework for Learning Spatio-Temporal Scene Graphs with Weak Supervision."_ ICLR. (2025).

---

### Official Review · Reviewer_z1So · 2025-06-24

**Clarity:** 2
**Significance:** 3
**Originality:** 3
**Rating:** 5
**Confidence:** 2

**Summary:**

The paper proposes a different learnable approach to probabilistic logic programming. Commonly for both prior and this work, the probability of a ground atom is defined by marginalizing over all possible words. In prior work the possible worlds include all possible similarity facts a \sim b, and the probability of each similarity fact is determined independently based on the similarity of embeddings for a and for b. Some sampled possible worlds can thus lack a proper transitive equivalence between constants. This work proposes to instead have every possible world defined as a well-defined equivalence relation. The distribution over such worlds is defined by a probability vector for each constant c that defines the probability of c belonging to each equivalence cluster. The paper discusses how to adapt inference engines for inference and learning with this new equivalence-based possible world distribution. Strong empirical results confirm the effectiveness of the proposed method.

**Questions:**

- Can you elaborate “even though in the global optimum we have P_f (y \sim z)=0.” in line 155?
- In the proposed approach every constants also has an embedding. Can you elaborate the difference in how embeddings are used in your work and in the prior similarity-based approaches?

**Ethical Concerns:**

["NO or VERY MINOR ethics concerns only"]

**Final Justification:**

I expect the final revision of the paper to address the clarity concerns that I expressed in my review.

**Limitations:**

I'd like to see more discussion on whether inference/learning becomes harder with the proposed approach compared to prior work.

**Paper Formatting Concerns:**

no concerns

**Quality:**

3

**Strengths And Weaknesses:**

The paper tackles an important issue in an original and principled way. I’m not an expert in logic programming, so some parts were very hard to follow. In particular, the crux of argument about the deficiencies of the embeddings-based in approach in lines 150-155 was lost on me. I’d like to see the notion of “global optimum” to be better explained there. Also it was not clear how much harder/easier the inference and learning are in the proposed approach compared to prior work.

Overall the exposition is very formal and abstract and hard to follow, but that is a common issue for logic programming papers.

---

> ### Author Rebuttal · Authors · 2025-07-30
>
> We sincerely thank the reviewer for their feedback and taking the time to review our paper.
>
> > In particular, the crux of argument about the deficiencies of the embeddings-based in approach in lines 150-155 was lost on me. I’d like to see the notion of “global optimum” to be better explained there. [...] Can you elaborate “even though in the global optimum we have P_f (y \sim z)=0.” in line 155?
>
> We have reworked Section 3.1 to improve clarity and accessibility to a broader audience. The overall goal of Section 3.1 is to show that using soft unification with embeddings is problematic, motivating the need for our new semantics. From the __optimization__ perspective, it is well-known that marginals of probabilistic logic (or indeed any distribution over discrete variables) are multilinear polynomials (see e.g. [1] or [2]). This implies all optima are global, making it relatively easy to optimize neurosymbolic models based on probabilistic logic. However, by using soft unification, the model is no longer multilinear in the parameters (e.g., the embeddings), and local optima may appear. This problem was already noticed by [3].  Example 7 was an indication of this, where following the gradient moves you away from the global optimum. We have replaced the old Example 7 with the following to demonstrate these ideas more explicitly, without needing a program with rules.
>
> _Example 7_. _Consider a soft unification program where embeddings are compared using a radial basis function, as done by the Neural Theorem Prover [4]. In other words, $P_f(a\approx b) =\phi(d(a,b))$, where $d(a,b)$ is the distance between the constants $a$ and $b$ in the embedding space and $\phi$ is a monotone function, typically a Gaussian.  Suppose now that we have a query whose probability is given by $(1 - P_f(a \approx b)) (1 - P_f(a \approx c))(1 - P_f(b \approx c)) + P_f(a \approx b) P_f(a \approx c)(1 - P_f(b \approx c)) + P_f(a \approx b)) P_f(b \approx c)(1 - P_f(a \approx c))$. Under the probabilistic semantics, the probabilities $P_f(\dots)$ are free parameters and hence this multilinear polynomial only has global optima. However, when soft unification is parameterized by embeddings, the probability of the query is computed by
> $(1-\phi(d(a,b))) (1-\phi(d(a,c)))(1-\phi(d(b, c)))+\phi(d(a, b)) \phi(d(a, c))(1-\phi(d(b , c)) + \phi(d(a, b))) \phi(d(b,c))(1-\phi(d(a, c)))$. This has a local maximum (e.g., using $\phi(x) = e^{-10x^2}$), because the vector space constrains the distances: $\vert d(a, b) - d(b, c)\rvert \leq  d(a, c) \leq d(a, b) + d(b, c)$._
>
> [1] Darwiche, Adnan. "A differential approach to inference in Bayesian networks." JACM (2003).
> [2] Roth, Dan, and Rajhans Samdani. "Learning multi-linear representations of distributions for efficient inference." Machine Learning (2009).
>
> > In the proposed approach every constant also has an embedding. Can you elaborate the difference in how embeddings are used in your work and in the prior similarity-based approaches?
>
> Both our work and previous research associate embeddings with symbols. However, we differ in the logic semantics associated with these embeddings. For example, what does it mean for the symbol $\mathtt{stop}$ of a logic formula $\mathtt{redlight} \rightarrow \mathtt{stop}$ to have an embedding $[0.130, 0.341, 0.023, …]$? Prior work, in particular, soft unification, compared symbols in the embedding space to assess how likely it is for two symbols to be "equivalent". On the other hand, we see embeddings as a latent random variable in a distribution over programs (see also Figure 1).
>
> > I'd like to see more discussion on whether inference/learning becomes harder with the proposed approach compared to prior work.
>
> We have added some more discussion on these questions in Section 5.
> (_On inference_) Firstly, from a basic complexity view, nothing changes: inference is still \#P-hard. Furthermore, elementary ground queries (such as whether two atoms are equal) remain tractable through our choice of factorization (c.f. Section 4.1). In practice, the most substantial difference of enforcing transitivity for exact inference is that the size of the ground program may increase. We partially mitigate this by using singularization and magic program transforms (c.f. Section 5).
> (_On Learning_) As optimization in our approach is multilinear and does not contain local optima, learning is easier compared to prior methods (c.f. the previous discussion about Sec. 3.1). For example, the NTP baseline uses 100 epochs for the experiments of Table 2, while we only need 1 epoch to converge.

---

> > ### Comment · Reviewer_z1So · 2025-08-08
> > **thank you**
> >
> > Thank you for your response. It seems like the final version of the paper will address my key concerns.

---

### Official Review · Reviewer_Qews · 2025-07-02

**Clarity:** 3
**Significance:** 3
**Originality:** 3
**Rating:** 5
**Confidence:** 3

**Summary:**

The paper formalizes a new approach for soft unification that is based on defining probability distributions over equivalences between symbols. Typically, in existing neurosymbolic methods soft unification is approximate since they are typically not true equivalences but on similarity (e.g. between symbol embeddings). The paper shows that under such situations, we may run the risk that inference in such models may derive facts that may not be true under equivalence. As a consequence, learning which is based on inference as a sub-step may be biased in such models. The paper adds equivalence semantics within probabilistic logic programming. Further, they show that a factorization of a distribution defined with the proposed framework can be realized through embeddings over the symbols. An exact inference pipeline is proposed for this model that uses optimizations based on symmetries along with a simple sampling-based approximation.

**Questions:**

Is there a reason why only transitivity was chosen within the experiments. Of course, it is true that transitivity occurs more naturally in real-world cases, but perhaps it would have helped to demonstrate enforcing other constraints.

If the embeddings are approximate (i.e. does not have dimensionality of the full vocabulary), does the proposed method reduce to other existing approximate approaches of soft unification using similarity?

**Ethical Concerns:**

["NO or VERY MINOR ethics concerns only"]

**Final Justification:**

The paper separates itself from other neuro-symbolic AI models by performing inference in the presence of exact equivalence as opposed to soft unification methods. This can have impacts in several application domains where we need to add domain knowledge that needs to be strictly enforced. The experiments show the benefit of this approach compared to existing neuro-symbolic methods that only allow approximate equivalences. Overall, this seems to be a good paper that merits acceptance.

**Limitations:**

Mentioned adequately.

**Quality:**

3

**Strengths And Weaknesses:**

Strengths
+ The paper formalizes novel probabilistic semantics for soft unification that is a central step in Neurosymbolic learning and inference.
+ The idea of equivalences/symmetries which has been used previously within relational models is applied in a novel setting that seems to be an interesting direction for Neurosymbolic AI.
+ The paper is well-written and rigorous in its approach.
+ The results show that when compared with soft unification that a lot of the other state-of-the-approaches typically perform, the probabilistic semantics over equivalences helps us enforce transitivity during learning which translates to better performance during inference.


Weaknesses
- While the proposed work is more general, the experiments enforce a single rule of transitivity.
- In more general cases, is the proposed learning method scalable? It seems like the singularization to be performed for exact weighted model counting will be computationally expensive.
- Regarding the comparison as such, it seemed to me like the proposed “exact” inference/learning method is being compared to approximate soft unification methods. In that sense, the approximate inference methods seem to be doing reasonably well it looked like in the graph benchmark at least.
- For the factorization, in the experiments, it is mentioned that the embedding dimension is as large as the vocabulary. This would seem like a limiting factor for many practical cases as such. What happens with a smaller embedding, I assume the distribution will not be expressive enough. But maybe this trade-off is important to address through some empirical results.

Overall, this seems to be a good, well-written paper that defines an important problem in Neurosymbolic models precisely and proposes novel probabilistic semantics for it.

---

> ### Author Rebuttal · Authors · 2025-07-30
>
> We sincerely thank the reviewer for taking the time to review our paper and providing useful feedback.
>
> > While the proposed work is more general, the experiments enforce a single rule of transitivity. [...] Is there a reason why only transitivity was chosen within the experiments. Of course, it is true that transitivity occurs more naturally in real-world cases, but perhaps it would have helped to demonstrate enforcing other constraints.
>
> The experiments use logic programs that may include dozens of rules, not just a transitive constraint (see Appendix B). Our work focuses on transitive equivalence because its importance has not yet been established in the context of differentiable proving and rule learning. Imposing such transitivity in an efficient fashion is nontrivial and required rethinking the prevailing probabilistic semantics of logic programming.
>
> > In more general cases, is the proposed learning method scalable? It seems like the singularization to be performed for exact weighted model counting will be computationally expensive.
>
> Singularization itself is a very efficient program transformation step that can be done in linear time in the number of rules. On the other hand, the weighted model counting step is \#P-hard, and hence faces scalability limitations. Notice that the latter is a challenge faced by every sufficiently expressive probabilistic programming language, and we have an approximation technique (c.f. lines 240-250) to keep the experiments reasonably fast.
>
> > Regarding the comparison as such, it seemed to me like the proposed “exact” inference/learning method is being compared to approximate soft unification methods. In that sense, the approximate inference methods seem to be doing reasonably well it looked like in the graph benchmark at least.
>
> We are not sure we understand this point. Both our experiments as well as all the baselines use approximate inference (c.f. lines 240-250 for our approximation strategy). Indeed, exact inference is \#P-hard in the size of the ground program (which can grow very quickly due to the use of templates in the experiments) and hence usually not feasible.
>
> > For the factorization, in the experiments, it is mentioned that the embedding dimension is as large as the vocabulary. This would seem like a limiting factor for many practical cases as such. What happens with a smaller embedding, I assume the distribution will not be expressive enough. But maybe this trade-off is important to address through some empirical results.
>
> Setting the embedding dimensionality to the vocabulary size can indeed become problematic for large programs. However, in these cases one can just take a smaller embedding dimension. If expressivity is a concern, the parameter count can be increased independently of vocabulary size with hierarchical mixtures without affecting tractability. In our experiments, the embedding dimensionality never was an issue, which is why we did not explore it further.
>
> > If the embeddings are approximate (i.e. does not have dimensionality of the full vocabulary), does the proposed method reduce to other existing approximate approaches of soft unification using similarity?
>
> No, it does not. The embeddings are only approximate in the sense that they limit the potential distributions over equivalence relations (in favour of tractability for certain queries). Further restricting expressivity by reducing the embedding dimensionality has no impact on the semantics. More specifically, transitivity is always guaranteed, and hence there cannot be a reduction to soft unification (which violates transitivity).

---

> > ### Comment · Reviewer_Qews · 2025-08-05
> >
> > Thanks for the detailed response. It clarified some misunderstandings I had.
> > About the exact vs approximate comparison, I think the point I was making was since all the other approaches use an approximate notion of enforcing transitivity, while the proposed approach uses an "exact" notion, is there any way to compare with other ways of enforcing "exact" transitivity (even on say smaller problems that are tractable). As an analogue, when comparing exact inference methods, we tend to compare with other exact inference solvers. But maybe this is not possible here which is fine.
> > For the issue of embeddings not being connected with the semantics, I think it would help to clarify this in the paper. Particularly since in practice we could plug-in different embeddings within the same approach, though the transitivity constraint is always guaranteed, what effect does the limitations in distributions over equivalences have is something that will be useful to consider.

---

> > > ### Author Response · Authors · 2025-08-06
> > >
> > > > I think the point I was making was since all the other approaches use an approximate notion of enforcing transitivity, while the proposed approach uses an "exact" notion, is there any way to compare with other ways of enforcing "exact" transitivity (even on say smaller problems that are tractable). As an analogue, when comparing exact inference methods, we tend to compare with other exact inference solvers.
> > >
> > > To the best of our knowledge, we are the first to investigate exact probabilistic inference in the presence of equivalence, in the sense that it respects both reflexivity, symmetry, and transitivity. If the reviewer is aware of an existing method that relies on this axiomatization of equivalence, we would be happy to compare against it.
> > >
> > > > For the issue of embeddings not being connected with the semantics, I think it would help to clarify this in the paper. Particularly since in practice we could plug-in different embeddings within the same approach, though the transitivity constraint is always guaranteed, what effect does the limitations in distributions over equivalences have is something that will be useful to consider.
> > >
> > > Perhaps we were not clear enough here. The embeddings are not completely disconnected from the semantics in the sense that our choice of semantics allows embeddings to be interpreted as factorizations of equivalence relations, which cannot be done for any arbitrary embedding spaces (c.f. Section 4.1). (So when we mentioned that transitivity is always guaranteed, this meant no matter the dimensionality of our embeddings, not for any other choice of embeddings.) Exploring more expressive distributions on equivalence relations could indeed be a productive avenue for future work.

---

### Official Review · Reviewer_FcSA · 2025-07-03

**Clarity:** 2
**Significance:** 2
**Originality:** 3
**Rating:** 5
**Confidence:** 3

**Summary:**

This paper argues that the use of embedding similarity in place of equivalence in neurosymbolic models is fundamentally problematic. That is, similarity violates transitivity, which ultimately hinders learning. In contrast to existing work on probabilistic programming which use a probability distribution over possible worlds (sets of facts), the authors introduce probability distributions over equivalences of constants. They propose using a latent variable to represent the equivalence distribution and make it tractable. Experiments on two tasks show consistent improvements over baseline methods.

**Questions:**

1. Where exactly am I misunderstanding section 3.1? While I am able to follow the proof of theorem 2 and believe I did my due diligence in checking the details, I fail to see how this supports the paper's main claim.

    - Would it be possible to include at least a summary of the proof of theorem 2 in the main text? I feel with the clarification I requested regarding this theorem, the proof would be helpful.

3. While the improvements of your method over the baselines in the experiments are an *indicator* that enforcing transitivity is helpful, I feel that additional experimentation or analysis would be helpful to support your claim. That is, just reporting higher AUCs is insufficient. Can you describe the extent to which each of these tasks depends on transitivity? Can you provide example failures of the other models that are caused by a lack of transitivity? Better yet, is it possible to, for example, quantify the number of links in experiment 1 that *require* rule learning in order to be predicted correctly?

**Ethical Concerns:**

["NO or VERY MINOR ethics concerns only"]

**Final Justification:**

Based upon the author rebuttal, I am confident that my concerns regarding clarity and accessibility of the paper will be resolved in the updated version.

**Limitations:**

Yes.

**Paper Formatting Concerns:**

127: I believe the x should be subscript.

134: Should $CA(R_{ex})$ be $C(R_{ex})$?

150: There is no $x$ predicate in the set of facts, only $y$ and $z$, but $x$ is used as a predicate on line 152. Is this correct? Unless $x$ is a variable that can be either $y$ or $z$. If so, this is a bit  confusing.

158: $f_+$ The + should be superscript.

Supplementary Section C: "Prossible" -> "Possible"

**Quality:**

2

**Strengths And Weaknesses:**

**Strengths**
 * The explanations and definitions  provided throughout, especially in the preliminaries where they are most needed, are very clear.
 * The factorization introduced in section 4.1 is quite clever.
 * The paper is overall very well-written and well-structured.


**Weaknesses**

* Section 3.1 was somewhat difficult for me to follow. This may be related to the notation confusion regarding $x$ in example 7, which I mention in the formatting concerns. I don't really see how theorem 2 supports the paper's main contribution of showing that soft unification does not adhere to the laws of equivalence. The proof supplied in the supplementary material did not offer clarification for me. What exactly is bad about requiring embeddings to be equal? Ultimately it is this section that drops my evaluation of the paper to "2: fair" for clarity and significance. Ensuring that this section is extremely clear to audiences with ranging understandings (i.e., not just experts) of neuro-symbolic methods, probabilistic logic programming, etc. would be a massive improvement.
  - I am unclear what exactly you are referring to by "nontransitive possible worlds" on line 156 and 161. Are they the "possible worlds not in an equivalence relation" mentioned on line 148? This makes Theorem 2 very unclear to me.
  - I assume a negative fact is a falsehood, e.g., $a \neq b$? This should be defined.
 * Similar to the above, I don't feel that the description of the experiments clearly address the research question "does transitivity matter".
 * The notation $e$ and $e'$ to denote equivalence relations in the Herbrand Universe and Base, respectively, was somewhat difficult for me to keep track of. Perhaps $e_u$ and $e_b$? This is a minor point and I understand that the authors may be using established notation here.

---

> ### Author Rebuttal · Authors · 2025-07-30
>
> We sincerely thank the reviewer for their feedback and taking the time to review our paper. We have split our answer into the two main points of concern.
>
> ## Concerning Section 3.1. (Equivalence vs Soft Unification)
>
> > Section 3.1 was somewhat difficult for me to follow. [...] Ensuring that this section is extremely clear to audiences with ranging understandings (i.e., not just experts) of neuro-symbolic methods, probabilistic logic programming, etc. would be a massive improvement.
>
> To address this concern, we reworked Section 3.1 to provide more discussion and make it more accessible. As we cannot upload a new pdf, we summarize the main points below before moving to the specific questions.
>
> The overall goal of Section 3.1 is to show that using soft unification with embeddings is problematic, motivating the need for our new semantics. First, there is the __optimization__ perspective. It is well-known that marginals of probabilistic logic (or indeed any distribution over discrete variables) are multilinear polynomials (see e.g. [1] or [2]). This implies all optima are global, making it relatively easy to optimize neurosymbolic models based on probabilistic logic. However, by using soft unification, the model is no longer multilinear in the parameters (e.g., the embeddings), and local optima may appear. This problem was already noticed by [3].  Example 7 was an indication of this, where following the gradient moves you away from the global optimum. We have replaced the old Example 7 with the following to demonstrate these ideas more explicitly, without needing a program with rules.
>
> _Example 7_. _Consider a soft unification program where embeddings are compared using a radial basis function, as done by the Neural Theorem Prover [4]. In other words, $P_f(a\approx b) =\phi(d(a,b))$, where $d(a,b)$ is the distance between the constants $a$ and $b$ in the embedding space and $\phi$ is a monotone function, typically a Gaussian.  Suppose now that we have a query whose probability is given by $(1 - P_f(a \approx b)) (1 - P_f(a \approx c))(1 - P_f(b \approx c)) + P_f(a \approx b) P_f(a \approx c)(1 - P_f(b \approx c)) + P_f(a \approx b)) P_f(b \approx c)(1 - P_f(a \approx c))$. Under the probabilistic semantics, the probabilities $P_f(\dots)$ are free parameters and hence this multilinear polynomial only has global optima. However, when soft unification is parameterized by embeddings, the probability of the query is computed by
> $(1-\phi(d(a,b))) (1-\phi(d(a,c)))(1-\phi(d(b, c)))+\phi(d(a, b)) \phi(d(a, c))(1-\phi(d(b , c)) + \phi(d(a, b))) \phi(d(b,c))(1-\phi(d(a, c)))$. This has a local maximum (e.g., using $\phi(x) = e^{-10x^2}$), because the vector space constrains the distances: $\vert d(a, b) - d(b, c)\rvert \leq  d(a, c) \leq d(a, b) + d(b, c)$._
>
> A second perspective is the __Bayesian__ one. Consider three symbols $a$, $b$, and $c$.
> There exists no embedding of those symbols that satisfies the following nontransitive probabilities: $P(\mathsf{a} \approx \mathsf{b}) = 1$, $P(\mathsf{b} \approx \mathsf{c}) = 1$, and $P(\mathsf{a} \approx \mathsf{c}) = 0$. Indeed, to satisfy the previous probabilities, the symbols $a$, $b$, and $c$ need to map to the same elements in the embedding space, implying an inherent transitivity (c.f. Theorem 2). Since, this nontransitive property cannot be represented by embeddings, by assigning mass to these probabilities, i.e., by setting $P(\mathsf{a}\approx\mathsf{b})=1$, $P(\mathsf{b} \approx \mathsf{c})=1$, and $P(\mathsf{a} \approx \mathsf{c})=0$, we leak probability mass, adversely impacting uncertainty quantification.
>
> [1] Darwiche, Adnan. "A differential approach to inference in Bayesian networks." _JACM_ (2003).
> [2] Roth, Dan, and Rajhans Samdani. "Learning multi-linear representations of distributions for efficient inference." _Machine Learning_ (2009).
> [3] de Jong, Michiel, and Fei Sha. "Neural theorem provers do not learn rules without exploration." (2019).
> [4] Rocktäschel, Tim, and Sebastian Riedel. "End-to-end differentiable proving." _NeurIPS_ (2017).
>
> > I don't really see how theorem 2 supports the paper's main contribution of showing that soft unification does not adhere to the laws of equivalence.
>
> The fact that soft unification does not adhere to transitivity (one of the laws of equivalence) is known and follows from its Definition (see lines 140-148). In contrast, Theorem 2 relates to _why_ it is problematic not to adhere to transitivity (see the previous discussion on the Bayesian perspective).
>
> > What exactly is bad about requiring embeddings to be equal?
>
> Requiring embeddings to be equal is definitely not a problem on its own. However, doing so on the probabilistic level without enforcing the same structure in the logic semantics is what leads to issues (see the previous discussion about Section 3.1).
>
> > I am unclear what exactly you are referring to by "nontransitive possible worlds" on line 156 and 161. Are they the "possible worlds not in an equivalence relation" mentioned on line 148? This makes Theorem 2 very unclear to me.
>
> With nontransitive possible world, we mean a possible world $w$ of the soft unification program $\mathcal{P}_{soft}$ where transitivity does not hold. For example, a possible world with $a \approx b$ and $ b \approx c $ but not $a \approx c$.
>
> > I assume a negative fact is a falsehood, e.g., $a \neq b$? This should be defined.
>
> Yes, we will define this in the text.
>
> > The notation $e$ and $e’$ to denote equivalence relations in the Herbrand Universe and Base, respectively, was somewhat difficult for me to keep track of.
>
> We will modify this notation to $e_\mathsf{U}$ and $e_\mathsf{B}$, respectively.
>
> ## Concerning Experiments on Transitivity
>
> > I don't feel that the description of the experiments clearly address the research question "does transitivity matter". [...] While the improvements of your method over the baselines in the experiments are an indicator that enforcing transitivity is helpful, I feel that additional experimentation or analysis would be helpful to support your claim. That is, just reporting higher AUCs is insufficient.
> Can you describe the extent to which each of these tasks depends on transitivity?
>
> The DeepSoftLog baseline is essentially an ablation of our method where transitivity is not enforced. So the (considerable) performance improvements over this baseline can be attributed to the addition of transitivity.
>
> > Can you provide example failures of the other models that are caused by a lack of transitivity?
>
> An example is in the differentiable finite state machine experiment, where the symbols represent states in an automaton. Consider the following transitions between states: $a\rightarrow b$, $c\rightarrow d$, and $e\rightarrow f$. Assuming we have that $b=c$ and $c=e$, it should be possible to transition $a\rightarrow b (=c) \rightarrow d$ and $a\rightarrow b (=e)\rightarrow f$. However, the latter derivation cannot be found without transitivity. This is the reason why DeepSoftLog often fails to optimize and has a large variance between experiments (c.f. Table 3).
>
> > Better yet, is it possible to, for example, quantify the number of links in experiment 1 that require rule learning in order to be predicted correctly?
>
> In the Countries (Table 1) and differentiable automata (Table 3) experiments, rules are necessary in each query. In other words, without rule learning, the performance would be near random. In the Nations knowledge graph (Table 2), this is less drastic – hence also the more modest performance improvements.
>
> > Paper Formatting Concerns
>
> Typos have been fixed. Regarding line 150, $x$ is an element of the Herbrand universe, while $X$ is a variable.

---

> > ### Comment · Reviewer_FcSA · 2025-08-01
> >
> > Your updated description in section 3.1 is much clearer now. Thank you. For me personally, the discussion of global vs. local optima was very helpful, specifically
> >
> > > It is well-known that marginals of probabilistic logic (or indeed any distribution over discrete variables) are multilinear polynomials (see e.g. [1] or [2]). This implies all optima are global, making it relatively easy to optimize neurosymbolic models based on probabilistic logic. However, by using soft unification, the model is no longer multilinear in the parameters (e.g., the embeddings), and local optima may appear. This problem was already noticed by [3]. Example 7 was an indication of this, where following the gradient moves you away from the global optimum.
> >
> > I would love to see this paragraph or some version of it included in the updated paper.
> >
> > ----
> >
> > Regarding the experiments, perhaps I could have been clearer: I trust that these datasets do indeed require rule learning, what I would have liked is a more detailed description of their structure to understand how/why rule learning is necessary. Currently, the paragraph at line 260 just states what the datasets are, leaving the reader to dig into the citations to get any sense of what kind of rules we're dealing with. The following part of your rebuttal starts to address this:
> >
> > > In the Countries (Table 1) and differentiable automata (Table 3) experiments, rules are necessary in each query. In other words, without rule learning, the performance would be near random. In the Nations knowledge graph (Table 2), this is less drastic – hence also the more modest performance improvements.
> >
> > That is, from your current experiment description and without being familiar with these datasets, I have no idea the extent to which rule learning is necessary. You here answer this about the Countries and automata data (i.e., 100% require rule learning), but if you can quantify what you mean by "less drastic" for Nations, that would be helpful for interpreting the results.
> >
> > ----
> >
> > > The DeepSoftLog baseline is essentially an ablation of our method where transitivity is not enforced. So the (considerable) performance improvements over this baseline can be attributed to the addition of transitivity.
> >
> > This is an important point. Please make sure to include this in the updated version.
> >
> > Thank you for the other clarifications.

---

> > > ### Author Response · Authors · 2025-08-07
> > >
> > > > I would love to see this paragraph or some version of it included in the updated paper. [...] This is an important point. Please make sure to include this in the updated version.
> > >
> > > We hope to include this using the extra page, in case of acceptance.
> > >
> > > > what I would have liked is a more detailed description of their structure to understand how/why rule learning is necessary. Currently, the paragraph at line 260 just states what the datasets are, leaving the reader to dig into the citations to get any sense of what kind of rules we're dealing with.
> > >
> > > Appendix B contains the rule templates that are used for each experiment, perhaps this is helpful to interpret what rule learning is happening. Furthermore, we added some examples of concrete learned rules below.
> > >
> > > Countries Benchmark:
> > > $\mathsf{locatedIn(X, Y) \leftarrow locatedIn(X, Z), locatedIn(Z, Y)}$
> > > $\mathsf{neighborOf(X, Y) \leftarrow neighborOf(Y, X)}$
> > >
> > > Nations Benchmark:
> > > $\mathsf{relbooktranslations(X, Y) \leftarrow students(X, Y)}$
> > > $\mathsf{releconomicaid(X, Y) \leftarrow militaryaliance(X,Y)}$
> > > $\mathsf{reldiplomacy(X, Y) \leftarrow officialvisits(X,Y)}$
> > >
> > > > I have no idea the extent to which rule learning is necessary. You here answer this about the Countries and automata data (i.e., 100% require rule learning), but if you can quantify what you mean by "less drastic" for Nations, that would be helpful for interpreting the results.
> > >
> > > For most knowledge graphs, pure embedding methods can achieve state-of-the-art results. So in this sense, the rule learning on the Nations knowledge graph is not really necessary. However, methods with symbolic rules, such as ours, have been investigated for advantages that extend beyond MRR, including improved interpretability and the ability to answer more complex queries than link prediction in a principled manner.

---

### Author Response · Authors · 2025-08-09
**Thank you**

We want to thank all the reviewers for their very constructive feedback. We will make sure all their comments are reflected in the new version of our work.

---

### Note · Authors · 2025-08-12

We thank all the reviewers. We summarize the main points that were raised, our responses, and the changes we will incorporate in the CR. **We will make sure our CR reflects all the feedback we got.**

**About Section 3.1.**
Our semantics is motivated by the fact that combining soft unification with embeddings is problematic from (1) an optimization and (2) a Bayesian perspective. Regarding (1), when using soft unification, the marginals induced by the corresponding probabilistic models are no longer multilinear in the embeddings and, hence, local optima may appear. The new version of Example 7 provided in our rebuttal exemplifies this point. This issue does not appear under our semantics. Regarding (2), probability mass is leaked from the program distribution when transitivity is violated. The example in our rebuttal explains this point.
We will use the example from the differentiable finite state machine experiments (discussed in our rebuttal) to explain the performance issues caused by the lack of transitivity.
In the CR, we will add the above and rework Section 3.1 as per the reviewers’ suggestions, e.g., we will explain Theorem 2.

**About the inference/learning cost of our approach.**
Singularization is a transformation that runs in linear time in the number of rules.
Inference is still #P-hard. To reduce inference overhead, we introduce an exact inference approach based on singularization and magic sets (Section 5).
About learning, as optimization in our approach does not contain local optima (see discussion above), learning is easier compared to SOTA. E.g., NTP needs 100 epochs for the experiments in Table 2; we need 1 epoch to converge.

**About the experiments.**
Our experiments use logic programs with hundreds of rules (Appendix B).
We ran additional experiments on the Kinship and UMLS benchmarks that showed the superiority of our approach against the SOTA.
The runtimes of our experiments are faster than those of baselines, such as DeepSoftLog, e.g., DeepSoftLog cannot handle the Nations benchmark due to scalability issues.
In the CR, we will include the above results and runtime measurements and rules that are learned during training in Appendix.

**About exact baselines.**
We are the first to investigate exact probabilistic inference in the presence of equivalence. The naive exact baseline would be a full axiomatization of logic equivalence, but this would be highly inefficient, making our experiments infeasible.

---

### Decision · Program_Chairs · 2025-09-17

**Decision:**

Accept (poster)

**Comment:**

Summary: This paper proposes a new probabilistic semantics for equivalence in logic programs, arguing that enforcing transitivity resolves key optimization issues in neurosymbolic learning. The proposed method demonstrates strong performance on several benchmarks.

Strengths: Reviewers unanimously agreed that the paper is well-written and well-motivated, tackling an important problem in neurosymbolic AI with a principled and clever technical approach.

Weaknesses: The primary initial concern raised by several reviewers was the lack of clarity in the section motivating the work, specifically regarding the negative consequences of violating transitivity in soft unification. It was well addressed during rebuttal.

Reasons for Decision: The paper has novel and sound technical contribution, strong empirical results. And the authors' rebuttal successfully clarified the core motivation and resolved the reviewers' main concerns about the pitfalls of non-transitive similarity in neurosymbolic models.

Discussion Process: The authors provided a revised explanation for their motivation, focusing on how soft unification introduces local optima, which convinced initially skeptical reviewers and led to a strong consensus for acceptance.